# Large language models are proficient in solving and creating emotional intelligence tests
Katja Schlegel [1,2] ✉, Nils R. Sommer [1] & Marcello Mortillaro [3]

Large Language Models (LLMs) demonstrate expertise across diverse domains, yet their capacity for emotional intelligence remains uncertain. This research examined whether LLMs can solve and generate performance-based emotional intelligence tests. Results showed that ChatGPT-4, ChatGPT-o1, Gemini 1.5 flash, Copilot 365, Claude 3.5 Haiku, and DeepSeek V3 outperformed humans on five standard emotional intelligence tests, achieving an average accuracy of 81%, compared to the 56% human average reported in the original validation studies. In a second step, ChatGPT-4 generated new test items for each emotional intelligence test. These new versions and the original tests were administered to human participants across five studies (total N = 467). Overall, original and ChatGPT-generated tests demonstrated statistically equivalent test difficulty. Perceived item clarity and realism, item content diversity, internal consistency, correlations with a vocabulary test, and correlations with an external ability emotional intelligence test were not statistically equivalent between original and ChatGPT-generated tests. However, all differences were smaller than Cohen's d ± 0.25, and none of the 95% confidence interval boundaries exceeded a medium effect size (d ± 0.50). Additionally, original and ChatGPT-generated tests were strongly correlated (r = 0.46). These findings suggest that LLMs can generate responses that are consistent with accurate knowledge about human emotions and their regulation.

Emotions are crucial for forming and maintaining social bonds, and effectively communicating them is vital for achieving positive outcomes in individuals and groups[1]. Thus, individuals with strong skills in recognizing, understanding, expressing, and responding to emotions (often summarized under the term ability emotional intelligence or ability EI[2]) often achieve better outcomes across different life domains, such as the workplace. For example, individuals with higher knowledge about emotions and emotion regulation strategies are perceived as warmer and more competent during workplace conflicts[3]. Conversely, poor emotional communication and management can lead to adverse outcomes, including loss of social support, impaired mental health, and group disintegration[1].

Based on such findings, the field of affective computing has set out to embed ability EI into machines and applications like chatbots and virtual assistants in order to enhance socio-emotional outcomes among their users. Since Rosalind Picard's seminal work in the late 1990s[4], affective computing and robotics have made remarkable progress, propelled by advancements in machine learning, neural networks, natural language processing, and other subdomains within artificial intelligence (AI). For example, automatic emotion recognition from video, audio, and text is now on par with human-level accuracy, even with naturalistic stimuli[5], and numerous applications to improve socio-emotional outcomes in healthcare, education, workplace, and other domains have been developed (for a review, see[6]). These include, for instance, socially assistive robots providing companionship and support[7], conversational agents enhancing the learning process in online educational settings by adapting to the emotional states of the learners[8], and tools that advise managers on how to improve workplace morale and productivity based on employee mood and well-being obtained from conversational surveys[9].

Despite these advances, however, the scope of many affective AI applications remains relatively narrow, with conversational agents often being limited to specific topics and lacking the ability to learn from and adapt to individual users[6,7]. To overcome these limitations, researchers have argued that the currently relatively independent subfields of affective computing—emotion recognition, generation, and application—need to be unified and more seamlessly integrated into AI systems to enable a more general affective AI that is applicable beyond isolated use cases[5].

[1]Institute of Psychology, University of Bern, Bern, Switzerland. [2]Institute of Psychology, Czech Academy of Sciences, Brno, Czech Republic. [3]Swiss Center for Affective Sciences, University of Geneva, Geneva, Switzerland. ✉e-mail: Katja.schlegel@unibe.ch

The recent rise of generative AI, particularly Large Language Models (LLMs) that power conversational agents like ChatGPT, may represent the critical advancement for achieving the goal of a general affective AI. These models exhibit human-like linguistic behavior, enabling real-time, sophisticated written conversations on any topic, making them promising candidates for artificial general intelligence (AGI) systems[10]. This development has opened up many exciting possibilities but also challenges, putting us at the outset of a "Brave new AI era"[11–14]. Importantly, state-of-the-art LLMs appear to generate responses consistent with knowledge of psychological concepts like personality, theory of mind, emotions, or empathy, despite not being explicitly trained with scientifically-based knowledge on these concepts[15–17]. As a result, ChatGPT (version 3.5), for example, responded to patients' medical questions in an online forum in a way that was rated significantly higher for both quality and empathy than human physicians[18].

The advent of widely accessible LLMs has sparked a lively debate about the scopes and limits of LLM-powered agents' human-like psychological capacities, such as whether ChatGPT and similar agents can truly convey empathy[10,19,20]. While this debate is important, especially regarding user acceptance of conversational agents or robots, a more fundamental and pragmatic question is how much LLMs' responses are consistent with accurate reasoning about emotions, their causes, consequences, associated expressions, and adaptive emotion regulation strategies. We argue that such reasoning, encapsulated in the construct of ability EI (e.g. refs. 2,21) and sometimes referred to as cognitive empathy[20], is a prerequisite for LLMs to be perceived as emotionally intelligent or empathic agents in settings such as healthcare, education, customer service, and other affect-laden interactions. Put differently, if LLMs fail to perform emotionally intelligent behaviors or tasks, they may lack the necessary prerequisites to achieve positive social outcomes or prevent detrimental ones in socio-emotional applications.

One straightforward way to address this question is to ask LLMs to solve performance-based tests from the realm of ability EI designed to measure such knowledge and abilities in humans and to compare LLMs' performance to human performance. In a study with an early version of ChatGPT (3.5), ChatGPT scored higher than the average human population on the Levels of Emotional Awareness Scale (LEAS)[22], in which test-takers are asked to write about how fictional characters in vignettes would likely feel[23]. This result suggested a more complex and nuanced processing of the vignettes and their emotional implications by ChatGPT compared to humans.

In the first part of the present research, we extend this promising early finding to a larger number of EI competencies, tests, and LLMs. Specifically, we compared the scores obtained by ChatGPT-4, ChatGPT-o1, Copilot 365, Claude 3.5 Haiku, DeepSeek V3, and Gemini 1.5 flash on five published ability EI tests to the average performance of human test-takers from the original validation studies. Two of the tests measure understanding of emotions and their causes and consequences by presenting vignettes describing emotional situations and asking test-takers to infer the most likely emotion or blend of emotions that the character in the scenario was feeling (Situational Test of Emotion Understanding, STEU[24]; Geneva EMOtion Knowledge Test, GEMOK-Blends[25]). The other three tests measure knowledge about the most appropriate course of action to regulate either one's own emotions (Geneva Emotional Competence Test, GECo—Emotion Regulation subtest[26]) or another person's emotions (GECo—Emotion Management subtest[26]; Situational Test of Emotion Management, STEM[24]). While all five tests use a situational judgment format with correct and incorrect response options, they differ substantially in scenario structure, setting (workplace or general life), complexity, length, and emotions included. This variety allows for more general conclusions about LLM's ability EI performance (see method section for details and example items in the supplementary material). Given the results in Elyoseph and colleagues' study[23] as well as results regarding related competencies such as Theory of Mind (e.g., false-belief tasks[10]), we expected all LLMs to generate significantly higher scores than the average of the human validation samples of each test.

In the second part of this research, we used ChatGPT-4 to create a new set of items (i.e., new scenarios and response options) for each of the five tests. We compared the psychometric quality of the ChatGPT-created test versions to that of the original tests through five studies conducted on Prolific. In each study, one of the new test versions (e.g., the ChatGPT-created version of the STEM) was administered alongside the original test version (e.g., the STEM), a vocabulary test to measure crystallized intelligence, and another ability EI test (e.g., the GECo Emotion Management subtest) to assess construct validity. Participants also rated the clarity, realism, and item content diversity for each new and original test version.

Compared to assessing LLMs' accuracy in solving ability EI test items, this part of the research aimed to more rigorously test the idea that ChatGPT-4, as one of the most widely used LLMs, is proficient at generating responses that demonstrate accurate knowledge about the structure and components of emotions and how they can be adaptively regulated in oneself and others. Because several studies have found the quality and believability of texts written by ChatGPT (in the context of simulating certain personality traits or cognitive abilities) to be satisfactory[27–29], we expected the ChatGPT-4-created test versions in our study to exhibit similar psychometric properties as the original test versions. Specifically, we expected that, across the five studies, original and ChatGPT-generated tests would show at most small differences in terms of test difficulty, internal consistency (Cronbach's alpha and average item-total correlations), ratings of clarity, realism, and item diversity, and their average correlation with crystallized intelligence and a different ability EI test.

## Methods
### Emotional intelligence tests
**Situational Test of Emotion Management (STEM)[24].** The STEM consists of 44 short (2–3 sentences) vignettes that each describe a fictional person experiencing a negative emotional situation (broadly reflecting anger, sadness, fear, or disgust) at work or in personal life. Participants are then asked to choose which of four actions would be the most effective for the person. The actions reflect six emotion regulation strategies (no regulation, situation selection, situation modification, attentional deployment, cognitive change, response modulation).

An example item is:

"Surbhi starts a new job where he doesn't know anyone and finds that no one is particularly friendly. What action would be the most effective for Surbhi? (a) Have fun with his friends outside of work hours. (b) Concentrate on doing his work well at the new job. (c) Make an effort to talk to people and be friendly himself. (d) Leave the job and find one with a better environment."

The full item list is available at https://doi.org/10.1037/a0012746.supp. Correct responses were defined by experts and responses were scored as 0 (incorrect) or 1 (correct) and aggregated into a total score reflecting the proportion of correct answers (possible range 0–1). The STEM was validated using a sample of 112 undergraduate students in Australia (Study 1[24]). This sample was a subset of the STEU validation sample ($N = 200$) described below.

**Situational Test of Emotion Understanding (STEU)[24].** The STEU consists of 42 vignettes (2–3 sentences each). In 36 items, the vignettes describe a concrete or abstract situation and participants choose which out of five emotion words best describes what the person involved is most likely to feel. An example item is: "A supervisor who is unpleasant to work for leaves Alfonso's work. Alfonso is most likely to feel? (a) Joy, (b) Hope, (c) Regret, (d) Relief, (e) Sadness".

In the remaining six items, an emotion is presented and participants choose what most likely happened to cause that emotion; for example: "Quan and his wife are talking about what happened to them that day. Something happened that caused Quan to feel surprised. What is most likely to have happened? (a) His wife talked a lot, which did not usually happen. (b) His wife talked about things that were different to what they usually discussed. (c) His wife told him that she might have some bad news. (d) His

wife told Quan some news that was not what he thought it would be. (e) His wife told a funny story." The full item list is available at https://doi.org/10.1037/a0012746.supp.

Correct answers were defined based on the emotion-specific appraisal patterns defined by appraisal theory[30]. Items were scored as correct or incorrect and aggregated into a total score reflecting the proportion of correct responses. The STEU was validated in a sample of 200 undergraduate students in Australia (68% women, age M = 21.1, SD = 5.6; Study 1[24]).

**Geneva EMOtion Knowledge Test—Blends (GEMOK-B)[25].** The GEMOK Blends consists of 20 vignettes (each about 100–140 words long) in which a fictional person experiences a situation characterized by two consecutive or blended emotions. The descriptions contain cues representing five emotion components (appraisals, feeling, action tendencies, expression, and physiology[31]). Based on these cues, participants are asked to infer which two emotion words best describe what the target person was feeling in the situation.

An example item is:

"Rachel is going to a concert of her favorite band with her best friends. Even before the concert starts, Rachel feels like singing and dancing. She chats and laughs with her friends, and enjoys the atmosphere. When the lead singer finally comes on stage and sings Rachel's favorite song, her heart starts to beat faster. Rachel closes her eyes so she can get totally absorbed in this moment. She wishes it could last forever. Which of the following emotions describe best what Rachel was experiencing during this episode? (a) Happiness and pleasure, (b) Joy and happiness, (c) Joy and pride, (d) pleasure and interest, (e) joy and sadness." The full item list is available at https://www.tandfonline.com/doi/suppl/10.1080/02699931.2017.1414687.

Correct answers were defined based on theoretically and empirically derived cue patterns for each emotion[32]. The GEMOK-Blends total score reflects the proportion of correct responses. The final version of the test was validated in 180 English-speaking MTurk workers (50% women; age M = 35.70, SD = 11.40).

**Geneva Emotional Competence Test in the workplace (GECo)[26].** The GECo consists of four subtests that measure emotion recognition ability from nonverbal cues, emotion understanding, emotion regulation in oneself, and emotion management in others. In the present study, only the latter two were used. The *Emotion Regulation* subtest consists of 28 vignettes (about four sentences each) describing situations in the workplace in which the test-taker is feeling a certain emotion. Participants are asked to choose two out of the four response options that best reflect what they would think in this situation. Two of the options correspond to adaptive emotion regulation strategies (acceptance, positive refocusing, focusing on planning, putting into perspective, or reappraisal), and two options correspond to maladaptive emotion regulation strategies (catastrophizing, rumination, self-blame, or other-blame[33]).

An example item of the GECo Regulation subtest is:

You successfully completed a very important project that took a lot of your time. When you return to your daily business, your boss tells you that he is unhappy that you neglected your other projects. You are very irritated about the lack of acknowledgement by your boss. What do you think? (a) You think that you should have been better organized and have worked on all the projects at the same time. (b) You think that you have to accept that bosses are never fully satisfied. (c) You think about the very positive feedback from the client for whom you completed the project. (d) You think that your boss is always unfair to you and that you should consider quitting if it continues.

Participants received zero points if they chose the two maladaptive options, 0.5 points if they chose one adaptive and one maladaptive option, and one point if they chose both adaptive options. These points are aggregated into a total score ranging from 0 to 1.

The *Emotion Management* subtest consists of 20 vignettes (about 4 sentences each) in which another person (colleague, client etc.)

experiences an emotion and participants are asked to choose, out of five courses of action, what they would most likely do. The five response options represent conflict management styles: accommodation, collaboration, compromise, competing, and avoidance[34].

An example item of the GECo Management subtest is:

"Your colleague with whom you get along very well tells you that he is getting dismissed and that you will be taking over his projects. While he is telling you the news he starts crying. He is very sad and desperate. You have a meeting coming up in 10 min. What do you do? (a) You take some time to listen to him until you get the impression he calmed down a bit, at risk of being late for your meeting; (b) You cancel your meeting, have a coffee with him and say that he has every right to be sad and desperate. You ask if there is anything you can do for him; (c) You tell him that you are sorry, but that your meeting is extremely important. You say that you may find some time another day this week to talk about it; (d) You suggest that he joins you for your meeting with your supervisor so that you can plan the transfer period together; (e) You cancel your meeting and offer him to start discussing the transfer of his projects immediately."

Correct responses are defined based on conflict management theory[34] which specifies the contextual factors under which each of the five strategies is the most appropriate one. Each strategy is the correct option in four of the 20 items. Correct responses are aggregated into a total score ranging from 0 to 1. The GECo Regulation and Management subtests were validated with English-speaking undergraduate students and university staff members (55% women; age M = 22.3, SD = 3.2; Study 2[26]).

**Assessing ChatGPT's performance in solving EI test items**
To test the assumption that LLMs would outperform humans in solving EI test items, ChatGPT-4, ChatGPT-o1, Gemini 1.5 flash, Copilot 365, DeepSeek V3, and Claude 3.5 Haiku were asked to solve the items of the STEM, STEU, GEMOK Blends, GECo Regulation and GECo Management subtests in December 2024 and January 2025. The LLMs received the original test instructions (i.e., to choose the correct response option) as well as all items that fit within the character limit (e.g., 10,000 characters for Copilot 365). The remaining items were inserted in separate prompts. Separate conversations were used for each test. Each LLM was prompted to solve each test 10 times in separate conversations. The number of correct responses for each test and trial was recorded, and for each LLM, the mean score and standard deviation across the 10 trials per test were computed. Mean scores across all six LLMs for each test were compared to the mean scores of human respondents obtained from the publications of the original validation studies using independent samples t tests. We preregistered the procedure for prompting ChatGPT-4 to solve each test once and comparing its performance with the human validation samples (link to preregistration: https://osf.io/mgqre/registrations). However, we did not preregister including other LLMs or to prompt each LLM 10 times (i.e., conducting ten trials).

**Assessing ChatGPT's performance in creating EI test items**
**Item generation with ChatGPT-4 and comparison with original item sets.** The first of the 10 trials of ChatGPT-4 in solving the five tests was used as a basis for item generation. First, for those items that were not correctly solved, we provided ChatGPT-4 with the correct answers. Second, we instructed ChatGPT-4 to generate the same number of new items and define their correct answers based on the items it had just solved (except for the STEM and STEU, where ChatGPT-4 was prompted to generate 18 and 19 items, respectively, akin to their validated short versions STEM-B and STEU-B[24]). Importantly, we attempted to have all new items created with just one prompt. Prompts were engineered in an iterative fashion based on an inspection of the answers provided by ChatGPT. The prompt for a given test was considered final when the generated items fulfilled the same formal criteria as the original test items (e.g., contained the desired strategies in the response options, the same emotions in the vignettes, etc.). The final prompts used for item generation, as well as the generated items, are provided in the Supplementary Material (Supplementary Notes 1 and 2).

**Similarity rating study.** This study (not preregistered) examined the degree of similarity between the original and ChatGPT-created test items for each of the five tests. Specifically, we aimed to test if some of the ChatGPT-created items were merely paraphrased versions of an original test item, which would challenge the idea that ChatGPT-4 is able to generate responses demonstrating accurate knowledge about emotional situations. To this end, 434 Prolific participants rated the similarity between all combinations of original and ChatGPT-created scenarios (i.e., the items without the response options) for each of the five tests (STEU, STEM, GEMOK, GECo Regulation, GECO Management). Participants were based in either the UK or the US and had indicated English as their first language. Gender and age were self-reported by participants (182 men, 243 women, 9 individuals with other gender identity; age $M = 38.5$ years, $SD = 12.7$ years).

For each of the five tests, parcels of about 50 randomly selected scenario pairs (each containing one original test item scenario and one ChatGPT-created item scenario) were created. For example, for the STEM, there were a total of 792 scenario pairs to be rated (44 original STEM items * 18 ChatGPT-created STEM items), which were randomly divided into 16 parcels of 49 or 50 scenario pairs. Across the five tests, a total of 3174 scenario pairs (divided into 64 parcels) were rated. Besides the scenario combinations, each parcel also contained three attention-check scenario pairs that were paraphrased versions of each other, the idea being that they should be rated as highly similar (i.e., with a 6 or 7). For example, one such attention-check scenario pair for the STEM was (1) Reece's friend points out that her young children seem to be developing more quickly than Reece's. Reece sees that this is true. [Original STEM scenario], (2) Rosie's friend observes that her young children appear to be developing faster than Rosie's, and Rosie realizes this is accurate. [Paraphrased STEM scenario].

Participants were randomly assigned to rate one of the parcels (i.e., containing about 50 scenario pairs plus 3 attention check pairs) and received the following instructions:

"You will now see pairs of scenarios describing emotional situations. Please rate how different or similar each pair of scenarios is, on a 7-point scale from "very different" to "very similar". For the purpose of this study, "very similar" scenarios are situations that are almost identical, but are described using different words and/or names.

Here is an example of two scenarios that we consider "very similar":
• Naya volunteers at an animal shelter and carefully plans her schedule each week. One morning, she arrives to discover her assigned tasks have been completely changed without any prior notice.
• Niya works at an animal shelter, organizing her schedule in advance each week. When she arrives one morning, she finds out her tasks were reassigned without her being informed.

In contrast, below is an example of two scenarios that we consider "very different":
• Riley ordered a new laptop for an important project, but it arrived damaged. Despite multiple calls to customer service, no resolution has been offered.
• At a big meeting, Malik accidentally projected a personal email instead of his presentation. He quickly tried to hide it, but everyone had already noticed."

On the next page, participants were asked to describe their task in the study using their own words in 1–2 sentences, before completing the ratings for all scenario pairs in their parcel. Depending on the duration of the survey (e.g., the GEMOK parcels took longer to read than the STEM parcels; average duration per survey 11–17 min), they were paid between 2 CHF and 2.50 CHF for their participation. Participants who did not correctly respond to the attention check items (i.e., did not rate all three attention check scenario pairs with 6 or 7) or wrote a nonsensical text when describing their task were excluded and are not part of the N of 434 described above. The study procedure resulted in 6 to 8 ratings for each scenario pair across all five tests. The average ratings for each scenario pair are provided in Supplementary Tables 1–8, with all pairs that received a rating of 5.0 and higher highlighted in blue. The raw data files can be accessed in the supplementary

**Table 1 | Distribution of highest similarity ratings for each of the 105 ChatGPT-generated scenarios**

| Highest similarity rating | Frequency | % | Cumulative % |
|---|---|---|---|
| 1.0 – 2.0 | 1 | 1.0 | 1.0 |
| 2.1 – 3.0 | 23 | 21.9 | 22.9 |
| 3.1 – 4.0 | 36 | 34.3 | 57.1 |
| 4.1 – 5.0 | 32 | 30.5 | 87.6 |
| 5.1 – 6.0 | 9 | 8.6 | 96.2 |
| 6.1 – 7.0 | 4 | 3.8 | 100.0 |

For each newly generated scenario, the value included in the Table represents the highest similarity rating observed across all comparisons with original test scenarios.

data folder in "data and analysis scripts" on OSF: https://osf.io/mgqre/files/osfstorage.

For each ChatGPT-created scenario (20 for GEMOK, 20 for GECo Emotion Management, 28 for GECo Regulation, 19 for STEU, and 18 for STEM, totaling 105 scenarios), we identified the original test scenario with the highest perceived similarity. For example, for item 1 from the ChatGPT-created STEU, the most similar original scenario was scenario 36, with a similarity rating of 4.4 (see column 1 in Supplementary Table 2). Table 1 presents the distribution of these highest similarity ratings across the 105 ChatGPT-created scenarios.

The numbers of scenarios created by ChatGPT that received a similarity rating of 5 or higher with at least one of the original scenarios were as follows: (1) STEU: 2 out of 19 scenarios, (2) STEM: 5 out of 18 scenarios, (3) GEMOK: 1 out of 20 scenarios, (4) GECo Regulation: 5 out of 28 scenarios, (5) GECo Management: 0 out of 20 scenarios. Overall, participants did not perceive a high level of similarity to any original test scenario in 88% of these newly created scenarios, while 12% of the scenarios created by ChatGPT across the five tests received a similarity rating of 5 or higher.

Table 2 shows the scenario texts for all scenario pairs with a similarity rating of 5.0 and higher. For the STEU, one of the original scenarios describes the appraisal structure for pride in an abstract way ("By their own actions, a person reaches a goal they wanted to reach"), while two ChatGPT-created scenarios present concrete situations in which a target person experiences either pride or satisfaction after achieving something (creating a piece of art, being rewarded for volunteer work).

For the STEM, one ChatGPT-generated item illustrates that an individual feels lonely after their colleague transfers to another company branch, and three scenarios from the original STEM also depict changes in a person's work context (e.g., losing touch with a former colleague). However, none of the original STEM scenarios mentions or describes loneliness. Additionally, two pairs of scenarios flagged as similar involve feelings of nervousness or fear in a job setting, though the specific situations differ (e.g., feeling nervous as an actor versus fearing to lead a team meeting). A similar pattern was observed in another two pairs of scenarios that revolve around comparable topics and/or emotions (two scenarios pertain to washing dishes/the kitchen, while two scenarios relate to the fear of flying, but the specific details vary).

For the GEMOK, the flagged scenario pair involves a similar setting (a target person watching their child's performance), but the development of the situation and the target emotions are distinctly different (joy/pride vs. sadness/pride). Lastly, in the GECo Regulation subtest, one newly created scenario and two original scenarios share a similar setting (obstacles to meeting a deadline), but the target emotions and described circumstances vary (e.g., anxiety vs. annoyance). Likewise, for the remaining four scenario pairs, while the settings are similar (starting a new job, facing pushback/criticism at work, not receiving a promotion), the target emotions and/or specific circumstances differ (e.g., sadness vs. worry vs. irritation).

Overall, as shown in Table 2, the scenario pairs perceived as similar relate to comparable settings or topics, yet they still exhibit distinct

differences in the specific situations described and/or in the emotions targeted. Therefore, we can conclude that ChatGPT-4 did not simply paraphrase the original test items when asked to create new scenarios and response options.

**General procedure of psychometric validation studies.** For each of the five tests, a separate online study was conducted on Prolific.com, where participants completed both the original test and the ChatGPT-generated version. For example, one study involved participants completing the original GEMOK-Blends and the ChatGPT-created GEMOK-Blends, while another study had a different sample complete the original STEU-B and the ChatGPT-generated STEU-B, and so on. Each sample also provided ratings of clarity and realism for both test versions, completed a card-sorting task, and took a vocabulary/crystallized intelligence test, as well as an additional test measuring the same EI dimension as the focal test (in the STEM-B study: GECo Management; in the STEU-B study: GEMOK Blends; in the GEMOK Blends study: STEU-B; in the GECo Regulation study: STEM-B; in the GECo Management study: STEM-B). The order of presentation for the original and ChatGPT-generated test versions, along with the associated clarity and realism ratings and the card-sorting task, was randomized. These were followed by the vocabulary test and the other EI test, which were always presented in this fixed order. Participants were not informed that one of the test versions had been generated by ChatGPT, and no references to LLMs or AI were made throughout the study. The studies were preregistered in October 2023 for the STEU, GECo Regulation, and GECo Management samples (https://osf.io/mgqre/registrations). The studies were approved by the ethics committee of the Faculty of Human Sciences at the first author's university (ID 20230803). Participants provided informed consent at the beginning of the study, and all relevant ethical regulations were followed.

**Samples.** All participants were native English speakers from the United States and UK recruited through Prolific.com with a generic description ("You will rate emotional situations on various criteria and complete a range of psychological questionnaires.") that did not refer to LLMs, AI, or emotional intelligence to reduce self-selection effects for participants. Participants were prevented from participating in more than one of the five studies, and participants were different from those recruited for the similarity rating study described earlier. The Ns, number of excluded participants (outliers), and demographic characteristics for each of the five samples are provided in Table 3. Participants self-reported their age, gender, highest level of education and their ethnicity (available in the data files). Past or current clinical diagnoses were not measured. Participants were excluded if they completed the survey in less than 15 min, scored 3 standard deviations or more below the mean on any of the included tests, or incorrectly responded to both attention-check items in the StuVoc (see below). Participants were paid based on study duration with an average compensation of £9 per hour.

**Instruments.** For GEMOK Blends, GECo Regulation, and GECo Management, see descriptions above. For STEU and STEM, the short forms STEU-B (19 items)[35] and STEM-B (18 items)[36] were used to reduce test-taking time. In the GEMOK Blends ChatGPT version, one item was presented to participants with the wrong response options and was therefore excluded from analyses, resulting in 19 items instead of 20.

After responding to each item of the original and ChatGPT-created tests, participants answered the following questions: "How plausible / realistic is this situation (including the response options)?" (Slider from 0 = "extremely implausible/ unrealistic" to 100 = "extremely plausible/ realistic") and "How clear is this situation (including the response options)?" (Slider from 0 = "extremely unclear/ confusing" to 100 = "extremely clear/ unambiguous"). The values on each of these questions were averaged across all items of the respective test version to form overall measures of realism and clarity.

After completing both test versions (original/ ChatGPT-generated), participants were presented with a list of all vignettes in that test (i.e., the original or the ChatGPT-created version) and read the following instructions ("card sorting task"): "Now the emotional situations you have just worked on are presented to you again. The situations are stacked on the left side of the screen. Please categorize the situations according to their content, putting similar situations together in one "pile". Create "piles" by dragging each situation into the boxes on the right. You can create up to 12 piles/ categories. Longer texts are abbreviated with "…". Please hover over the text with your mouse to read the full situation before deciding what "pile" to put it on. Create as few piles as possible, but as many as seem right to you." The average number of piles created was used as an index of diversity of the scenarios/ vignettes. A smaller number of piles created by participants indicates that the vignettes are more similar in content, whereas a higher number of piles indicates more variety and diversity of situations covered in the vignettes. Due to an error (24 piles were provided instead of 12), for the GECo Emotion Regulation subtest (original version), 10 participants who had created more than 12 piles were excluded when calculating the number of categories, yielding N = 85 for the category score.

Participants also completed a short version of the StuVoc1 vocabulary test[37] which taps into crystallized intelligence. In this test, participants are presented with words and example sentences containing the word and are asked to choose which out of four options correctly describes the meaning of each word in the corresponding sentence. One example item is: "What is the meaning of the word ROUBLE? "He had a lot of ROUBLES." (a) Russian money, (b) very valuable red stones, (c) distant members of his family, (d) moral or other difficulties in the mind". Based on the item difficulties and item discrimination indices of the 50 StuVoc1 items[37], we created a 20-item short version by selecting items with an item-total-correlation above r = 0.29, sorting these items by item difficulty, and choosing every other item in this list. The reliability of the 20-item version was good, with Cronbach's alphas ranging from 0.70 to 0.84 (mean alpha across the five studies = 0.80). In addition to the 20 items, we administered two easy items of the same format recommended by Vermeiren and colleagues as attention check items[37].

**Power analysis.** A priori power analyses were conducted with G*Power[38] to determine the necessary sample size for each of the five studies (one study per EI test). For mean comparisons (test scores, clarity and realism ratings, and card-sorting categories), we set a medium effect size ($d = 0.50$), and for correlations (test relationships with intelligence and another EI test), we used $r = 0.30$, corresponding to guidelines for moderate effects[39]. Power was set at 0.80, with $\alpha = 0.05$. The power analysis indicated that the required sample size per study was $N = 34$ for mean comparisons and $N = 82$ for correlation analyses. We selected medium effect sizes for the individual studies because we planned to assess the main hypotheses regarding the similarity of the original and ChatGPT-generated EI tests based on the aggregated results across the five studies, where the combined sample size would be sufficiently large to detect smaller effects.

For the combined sample ($N = 467$), we conducted power analyses for equivalence tests using the TOSTER R package[40]. For the analysis of mean differences in test scores, clarity and realism ratings, and the number of categories in the card-sorting task, we predefined a smallest effect size of interest (SESOI) of $d = \pm 0.20$, corresponding to a small effect size[39]. While we were unable to identify prior studies establishing a meaningful difference for these measures, we considered a difference of ~0.20 SD to be a minimally noticeable effect in test validation contexts. A power analysis using the power_t_TOST function showed that the combined sample ($N = 467$) had 99.2% power to detect equivalence within the bounds of $d = \pm 0.20$ ($\alpha = 0.05$).

For the comparison of correlations (examining whether the original and GPT-generated tests differed in their relationships with intelligence and another ability EI test), we predefined a SESOI of $r = \pm 0.15$. This threshold was based on a meta-analysis[41] which reported 95% confidence intervals for

## Table 2 | Original and ChatGPT-created scenarios with a similarity rating of >5.0

| Test | Similarity rating | ChatGPT-created scenario | Original scenario |
|---|---|---|---|
| STEU | 6.5 | 2: Chris felt a wave of contentment as he gazed at the artwork he had spent weeks perfecting. | 24: By their own actions, a person reaches a goal they wanted to reach. |
| STEU | 5.5 | 15: Olivia experienced a sense of fulfillment seeing the bright smiles of the children she had volunteered to assist. | 24: By their own actions, a person reaches a goal they wanted to reach. |
| STEM | 6.4 | 1: Karen's favorite coworker, Sam, has been transferred to another branch, leaving Karen feeling quite lonely. | 5: Wai-Hin and Connie have shared an office for years but Wai-Hin gets a new job and Connie loses contact with her. |
| STEM | 5.7 | 1: Karen's favorite coworker, Sam, has been transferred to another branch, leaving Karen feeling quite lonely. | 32: Mallory moves from a small company to a very large one, where there is little personal contact, which she misses. |
| STEM | 5.3 | 1: Karen's favorite coworker, Sam, has been transferred to another branch, leaving Karen feeling quite lonely. | 34: Blair and Flynn usually go to a cafe after the working week and chat about what's going on in the company. After Blair's job is moved to a different section in the company, he stops coming to the cafe. Flynn misses these Friday talks. |
| STEM | 5.9 | 3: Laura is nervous about an upcoming presentation she has to give in front of the company board. | 10: Darla is nervous about presenting her work to a group of seniors who might not understand it, as they don't know much about her area. |
| STEM | 5.3 | 3: Laura is nervous about an upcoming presentation she has to give in front of the company board. | 30: Billy is nervous about acting a scene when there are a lot of very experienced actors in the crowd. |
| STEM | 5.7 | 10: Olivia fears public speaking and is asked to lead a team meeting. | 30: Billy is nervous about acting a scene when there are a lot of very experienced actors in the crowd. |
| STEM | 5.7 | 11: Max's roommate consistently leaves dirty dishes around the house. | 22: Evan's housemate cooked food late at night and left a huge mess in the kitchen that Evan discovered at breakfast. |
| STEM | 5.6 | 14: Susan is anxious about flying and has a trip coming up. | 6: Martina is accepted for a highly sought after contract, but has to fly to the location. Martina has a phobia of flying. |
| GEM-OK | 5.3 | 2: Ben watches his son's piano recital. The auditorium is filled with expectant parents and well-wishers. His son begins with confidence but stumbles on some notes midway. Ben's heart skips a beat, memories of his own failures as a musician flooding back. However, his son regains composure, finishing with a flourish, earning applause from the audience. | 2: Robert's six-year old daughter is participating in a show of her ice-skating school for the first time. When Robert sees her appear on the ice, he smiles widely, his heartbeat quickens and he feels like jumping up from his seat. He feels so good he wants the show to go on forever. During her short solo part, Robert tells his neighbor excitedly in a loud voice that his daughter is performing. He feels like showing off and telling everybody around him about his daughter. |
| GECo Regulation | 5.2 | 2: An unforeseen complication arises in your project. The initial timeline, which was already tight, now seems impossible. Waves of anxiety rush over you as the looming deadline approaches. | 3[a]: You are annoyed because your supervisor reminds you of tomorrow's deadline although other people are delaying the work. |
| GECo Regulation | 5.7 | 2: An unforeseen complication arises in your project. The initial timeline, which was already tight, now seems impossible. Waves of anxiety rush over you as the looming deadline approaches. | 12[a]: You are worried because of unexpected technical issues with a new IT system. |
| GECo Regulation | 6.1 | 9: You've recently been promoted. With new responsibilities, you find yourself struggling to keep up. Colleagues you once considered friends seem distant, making you feel isolated and sad. | 10[a]: You are worried that you may not meet the expectations at your new job. |
| GECo Regulation | 5.1 | 11: You present a new, innovative approach to a recurring problem in a meeting. However, it's met with immediate resistance from a senior member, causing public embarrassment and anger. | 7[a]: You are annoyed because your colleague points out a mistake you made during a client meeting. |
| GECo Regulation | 5.6 | 16: During a video call, a colleague unexpectedly challenges your solution. Their unexpected opposition throws you into a state of irritation. | 7[a]: You are annoyed because your colleague points out a mistake you made during a client meeting. |
| GECo Regulation | 6.7 | 23: A promotion you had been eyeing goes to a less experienced colleague. This unexpected decision instills feelings of despair and confusion. | 23[a]: You are sad because your colleague is promoted and becomes your new supervisor. |

[a]For the GECo Regulation subtest, the original items are protected by copyright and cannot be printed here; the texts show summaries of each item. Complete similarity ratings can be found in Supplementary Tables 1–8.

**Table 3 | Sample descriptions of the five validation studies**

| Sample | Removed outliers | Final N (after removing outliers) | Age M (SD) | Self-reported gender | | | Self-reported ethnicity | | | | | |
| --- | --- | --- | --- | --- | --- | --- | --- | --- | --- | --- | --- | --- |
| | | | | N women | N men | N other[a] | N White or Caucasian | N Black or African American | N Asian or Pacific Islander | N Hispanic or Latino | N Multiple ethnicity or other | N Prefer not to say |
| STEM-B | 1 | 90 | 39.4 (10.0) | 68 | 21 | 1 | 76 | 3 | 5 | 0 | 5 | 1 |
| STEU-B | 3 | 94 | 37.1 (10.5) | 46 | 48 | 0 | 77 | 7 | 3 | 0 | 6 | 1 |
| GEMOK Blends | 2 | 91 | 35.5 (10.5) | 44 | 46 | 1 | 71 | 6 | 8 | 1 | 3 | 2 |
| GECo Regulation | 0 | 95 | 36.9 (11.8) | 47 | 48 | 0 | 78 | 7 | 6 | 0 | 3 | 1 |
| GECo Management | 1 | 97 | 38.5 (11.3) | 50 | 47 | 0 | 79 | 9 | 5 | 1 | 2 | 1 |

[a]Participants reported a different gender identity or chose the option "prefer not to say".

correlations between intelligence and different ability EI branches, ranging in width from $r = 0.08$ to $r = 0.35$ (Table 2[41]). We adopted the average width of these confidence intervals ($r = 0.15$) as the SESOI, as differences smaller than ±0.15 would be within the range of expected variability in EI-intelligence correlations. A power analysis using the power_z_cor function showed that the combined sample ($N = 467$) had 89.3% power to detect equivalence within these bounds ($\alpha = 0.05$). Given this sufficient power, and in the absence of prior literature guiding the choice of a SESOI for item-total correlations, we applied the same SESOI ($r = ±0.15$) for comparing the average item-total correlations between the original and ChatGPT-generated test versions.

## Results
### Assessing LLM performance in solving EI test items
As shown in Table 4, as expected, all tested LLMs achieved a higher proportion of correct responses across all five tests compared to the mean scores of the human validation samples published by the original test authors (mean accuracy across all LLMs was 81% versus 56% among the human samples). Notably, all LLMs performed more than one standard deviation above the human mean, with ChatGPT-o1 and DeepSeek V3 exceeding two standard deviations above the human mean. The LLMs also exceeded human performance in each of the five EI tests individually, with large effect sizes (see Table 5).

There was substantial agreement among the six LLMs, with the Intraclass Correlation (ICC) across all 105 test items being.88. To further examine similarities and differences between human test takers and the six LLMs when solving the five EI tests, we calculated correlations across all 105 test items between the proportions of correct responses in the human samples (i.e., the mean scores on each item) and the proportions of correct responses among the six LLMs (not preregistered). The datafile for this calculation is called "comparison_humans_llm_osf.xlsx" and can be accessed in the supplementary data folder in "data and analysis scripts" on OSF: https://osf.io/mgqre/files/osfstorage. Across the 105 items, the correlation between human and LLM scores was $r = 0.46$, indicating that items with higher proportions of correct responses among humans (i.e., easier items) were also more frequently solved correctly by the LLMs. Detailed item-level comparisons are described in the Supplementary Notes 5 (p. 59 of the supplementary material).

### Assessing ChatGPT's performance in creating EI test items
To evaluate whether LLMs generate test items with psychometric properties comparable to existing ability EI tests, we proceeded in four steps: First, we performed $t$ tests on the pooled data across the five separate studies ($N = 467$) for test difficulty (proportion of correctly solved items by the human sample), clarity and realism ratings, and item content diversity (measured as the average number of categories in which participants sorted the item scenarios from each test version), conducted using SPSS version 27, as well as multilevel meta-analyses for internal consistency and construct validity (correlations with the vocabulary test and the other ability EI test included in each study) conducted in R version 4.4.1. Because the assessed outcome variables captured largely independent constructs (e.g., test difficulty was unrelated to clarity ratings), and because the hypotheses for all variables were preregistered, no correction for multiple comparisons was applied. Second, when results were not statistically significant, we performed equivalence tests using the Two One-Sided Tests (TOST) procedure implemented in the TOSTER package in R[40] to examine whether original and ChatGPT-created tests were statistically equivalent regarding a given psychometric characteristic (e.g., in their mean scores). Third, when equivalence tests were not significant (meaning that statistical equivalence between original and ChatGPT-created versions could not be established), we still considered original and ChatGPT-created test versions to be similar regarding a given psychometric characteristic when the 95% CI of the effect obtained in the $t$ tests or meta-analyses was within the range of small effects, defined as Cohen's $d$ not or only marginally exceeding ±0.20, or Pearson's $r$ not exceeding ±0.15.

**Table 4 | Means and standard deviations of test scores achieved by LLMs**

|  | ChatGPT-4 | ChatGPT-o1 | Copilot 365 | Claude 3.5 Haiku | Gemini 1.5 flash | DeepSeek V3 | LLM total |
|---|---|---|---|---|---|---|---|
| STEM | 0.75 (0.03) | 0.83 (0.03) | 0.79 (0.03) | 0.75 (0.06) | 0.76 (0.03) | 0.80 (0.01) | 0.78 (0.05) |
| STEU | 0.72 (0.02) | 0.85 (0.01) | 0.78 (0.04) | 0.77 (0.02) | 0.77 (0.04) | 0.81 (0.02) | 0.78 (0.05) |
| GEMOK-Blends | 0.80 (0.03) | 0.87 (0.05) | 0.85 (0.03) | 0.87 (0.02) | 0.83 (0.05) | 0.90 (0.00) | 0.85 (0.05) |
| GECo Regulation | 0.87 (0.01) | 0.89 (0.01) | 0.92 (0.02) | 0.89 (0.03) | 0.78 (0.04) | 0.82 (0.02) | 0.86 (0.05) |
| GECo Management | 0.82 (0.05) | 0.74 (0.03) | 0.75 (0.05) | 0.68 (0.07) | 0.67 (0.03) | 0.85 (0.00) | 0.75 (0.08) |
| Mean scores | 0.79 (0.02) | 0.84 (0.01) | 0.82 (0.01) | 0.79 (0.03) | 0.76 (0.02) | 0.84 (0.01) | 0.81 (0.03) |

For each LLM, the displayed values are the means and standard deviations of 10 independently repeated testing trials. All LLM data was collected in December 2024/ January 2025.

**Table 5 | T-tests comparing LLMs and human validation study samples**

|  | LLM total | Human sample | Human validation sample source | t test | Cohen's d |
|---|---|---|---|---|---|
| STEM | 0.78 (0.05) | 0.52 (0.07) | Study 1[24]; N = 112 | t(170) = 25.483, p < 0.001 | 4.077 [3.544; 4.610] |
| STEU | 0.78 (0.05) | 0.60 (0.13) | Study 1[24]; N = 200 | t(258) = 10.483, p < 0.001 | 1.543 [1.226: 1.861] |
| GEMOK-Blends | 0.85 (0.05) | 0.67 (0.18) | Study 2[25]; N = 180 | t(238) = 7.639, p < 0.001 | 1.139 [0.829: 1.448] |
| GECo Regulation | 0.86 (0.05) | 0.56 (0.11) | Study 1[26]; N = 149 | t(207) = 20.276, p < 0.001 | 3.100 [2.678: 3.522] |
| GECo Management | 0.75 (0.08) | 0.45 (0.18) | Study 1[26]; N = 149 | t(207) = 12.412, p < 0.001 | 1.898 [1.547; 2.248] |

t tests remained statistically significant after applying the False Discovery Rate (FDR) correction[42], with all p values remaining below 0.001.

In the fourth step, we compared the original and ChatGPT-4-created versions of each individual test on the same outcome variables: test difficulty, clarity and realism, item content diversity, Cronbach's alpha, correlations with the vocabulary test (StuVoc), and correlations with the other ability EI test in the study to assess construct validity. For these test-level analyses, p-values for each outcome variable were corrected for multiple comparisons using the False Discovery Rate (FDR) correction[42]. All statistical tests were two-sided.

Histograms for difference scores in the proportion of correct responses, clarity, realism, and number of categories between the original and ChatGPT-created versions suggest that all difference scores were approximately normally distributed and had very few extreme outliers; however, data distribution was not formally tested. Participant gender was not included in the analyses as it was not expected to affect differences between original and ChatGPT-created test versions. Item-level analyses, including mean scores, item-total correlations, clarity, and realism ratings for all original and ChatGPT-generated items, are provided in the Supplementary Material (Supplementary Notes 4, pp. 48–58).

For test difficulty (see Table 6), the t test on the pooled dataset was not significant, t(466) = −1.223, p = 0.222. We then performed an equivalence test with the t_TOST function in TOSTER[40] with a predefined smallest effect size of interest (SESOI) of d = ± 0.20, corresponding to a small effect size[39]. We considered a difference of ~0.20 SD to be a minimally noticeable effect in test validation contexts. Results indicated that the difference between the original and GPT-generated test scores was statistically equivalent within the bounds of d ± 0.20, t(466) = 19.61, p < 0.001. In addition, Cohen's d in the t test was very small, with the 95% CI being within our predefined bounds (d = −0.057 [−0.147; 0.034]). On the level of individual tests, results showed that the ChatGPT-created versions were significantly easier for the STEM and GECo Regulation (indexed by higher mean scores), whereas the original test versions were easier for the GEMOK-Blends and GECo Management (Table 6).

For clarity ratings (Table 6), the mean difference was not statistically significant, t(466) = −1.799, p = 0.073, and the equivalence test for a SESOI of d ± 0.20 was not significant either, t(466) = −1.50, p = 0.93, indicating that we could not conclude that clarity ratings were statistically equivalent between the original and ChatGPT-created versions. However, Cohen's d was very small and the 95% CI was within our predefined boundaries

(−0.083 [−0.174; 0.008]). Results for the individual EI tests indicated that clarity was rated significantly higher in the ChatGPT-created versions for all tests except the GEMOK Blends.

For realism ratings (Table 7), the mean difference was statistically significant, t(466) = −2.746, p = 0.006, with ChatGPT-generated tests obtaining a slightly higher average compared to the original test versions. Again, Cohen's d was very small (−0.127 [−0.218; −0.036]), with the lower CI boundary slightly exceeding the predefined boundaries of a small effect. On the level of individual tests, realism was rated as significantly higher for the ChatGPT-generated versions of the STEU-B, GECo Regulation, and GECo Management subtests; and significantly higher for the original version of the GEMOK Blends.

In the card sorting task (Table 7), participants overall used significantly more categories when sorting the original scenarios compared to the ChatGPT-generated scenarios, t(456) = 4.446, p < 0.001, suggesting they were perceived as more diverse in content. Cohen's d was small, but the CI exceeded our predefined boundaries (0.208 [0.115; 0.301]). On the individual test level, significant differences were found for the GEMOK Blends, GECo Regulation, and GECo Management items, with the original versions being perceived as more diverse in content.

To compare internal consistency across all tests (Table 8), a fixed effects multilevel meta-analysis was conducted on the Fisher-z-transformed average item-total correlations of each test, with test type (original vs. ChatGPT-generated) as a moderator. Average item-total correlations were used instead of Cronbach's alpha because alphas cannot be directly meta-analyzed. The analysis was conducted with the metafor R package with the restricted maximum likelihood (REML) method in the R package "metafor"[43]. QM is the test statistic for the moderator variable "original" vs "GPT-created" when the intercept is included, and was not significant (QM = 0.635; df = 1; p = 0.426), indicating that test type did not significantly moderate the average item-total correlation. An equivalence analysis on the average item-total correlations (for original tests: r = 0.183; for ChatGPT-generated tests: r = 0.259) was then conducted with the compare_cor function in TOSTER; z = −0.071, p = 0.139. The non-significant result indicated that the two correlations were not statistically equivalent within the predefined bounds of r ±0.15 (see section on power analysis). The effect size of the difference was small (d = −0.152 [−0.547; 0.223]), but the CI exceeded our predefined boundaries. Regarding individual tests, Cronbach's

**Table 6 | Test scores and clarity ratings for the original and ChatGPT-4-created test versions**

| Study | Test score means (0–1) | | | | Clarity ratings (0–100) | | | |
|---|---|---|---|---|---|---|---|---|
| | Original | GPT | t test | Cohen's d [95% CI] | Original | GPT | t test | Cohen's d [95% CI] |
| STEM-B | 0.69 (0.14) | **0.93** (0.07) | t(89) = −17.093, p < 0.001 | −1.802 [−2.135; −1.464] | 84.8 (14.7) | 89.1 (11.1) | t(89) = −4.292; p = 0.002 | −0.452 [−0.668; −0.234] |
| STEU-B | 0.63 (0.13) | 0.62 (0.10) | t(93) = 1.116, p = 0.267 | 0.115 [0.088; 0.318] | 76.4 (17.8) | **84.9** (12.2) | t(93) = −7.946; p = 0.002 | −1.060 [−10.526; −6.317] |
| GEMOK-Blends | **0.68** (0.14) | 0.47 (0.12) | t(90) = 12.523, p < 0.001 | 1.313 [1.030; 1.592] | **86.0** (11.04) | 73.6 (17.4) | t(90) = 8.129, p = 0.002 | 0.852 [0.610; 1.091] |
| GECo Regulation | 0.54 (0.12) | **0.65** (0.21) | t(94) = −7.424, p < 0.001 | −0.762 [−0.989; −0.532] | 80.6 (15.7) | **82.6** (15.3) | t(94) = −2.335; p = 0.022 | −0.240 [−0.443; −0.035] |
| GECo Management | **0.54** (0.15) | 0.49 (0.13) | t(96) = 3.886, p < 0.001 | 0.395 [0.187; 0.600] | 77.7 (17.1) | **80.3** (15.1) | r(96) = −2.467, p = 0.019 | −0.250 [−0.452; −0.048] |
| Pooled dataset | 0.62 (0.15) | 0.63 (0.21) | t(466) = −1.223, p = 0.222 | −0.057 [−0.147; 0.034] | 81.0 (15.9) | 82.1 (15.2) | t(466) = −1.799, p = 0.073 | −0.083 [−0.174; 0.008] |

Values in boldface indicate a significantly higher value compared to the other test version. Displayed p values for the individual studies are FDR-corrected.

alpha was significantly higher for the original STEU version than for the ChatGPT-created version and significantly higher for the ChatGPT-created GECo Regulation version than the original (Table 8).

The original and ChatGPT-created versions were significantly positively correlated, with a large effect size (mean $r$ weighted by sample size = 0.46, $p < 0.001$; Table 8), suggesting that they measure similar constructs.

To compare original and ChatGPT-generated tests in their correlations with the StuVoc (vocabulary test), a fixed effects multilevel meta-analysis was conducted on the Fisher-z-transformed correlations of each test, with test type (original vs. ChatGPT-generated) as a moderator (Table 9). The result was not significant, $QM = 2.651$; $df = 1$; $p = 0.104$. An equivalence test with the predefined SESOI of $r \pm 0.15$ (see section on power analysis) was then conducted on the average StuVoc correlations with original ($r = 0.244$) and ChatGPT-generated ($r = 0.137$) tests, $z = -0.047$, $p = 0.236$, suggesting that the correlations with StuVoc were not equivalent between original and ChatGPT-generated tests. In addition, while Cohen's $d$ was small, the CI exceeded our predefined boundaries (0.217 [−0.044; 0.492;]). These findings suggest that while the difference in correlations was small, it could not be ruled out that the ChatGPT-generated tests had a meaningfully weaker association with StuVoc than the original tests. For each test individually, correlations with the StuVoc did not differ significantly between the original and ChatGPT-created versions (Table 9). However, post-hoc power analyses with G*Power revealed that these individual analyses were underpowered. For instance, with $N = 97$ and $\alpha = 0.05$, the power to detect a true correlation difference of $r = 0.16$ between the two GECo Management test versions was only 35%.

Finally, to compare original and ChatGPT-generated versions regarding their correlations with another ability EI test (see Table 6 for the name of the other test administered in each study), a fixed effects multilevel meta-analysis was conducted on the Fisher-z-transformed correlations of each test, with test type (original vs. ChatGPT-generated) as a moderator (Table 9). The result was not significant, $QM = 2.189$; $df = 1$; $p = 0.149$. An equivalence test with the predefined SESOI of $r \pm 0.15$ was then conducted on the average ability EI correlations with original ($r = 0.323$) and ChatGPT-generated ($r = 0.236$) tests, $z = -0.064$, $p = 0.164$, suggesting that the correlations with another ability EI test were not equivalent between original and ChatGPT-generated tests. Again, while Cohen's $d$ was small, the CI exceeded our predefined boundaries (0.197 [−0.064; 0.471]). These findings suggest that while the difference in correlations was small, it could not be ruled out that the ChatGPT-generated tests had a meaningfully weaker association with other ability EI tests than the original tests. For each test individually, correlations with the other ability EI test did not differ significantly between the original and ChatGPT-created versions (Table 9), but as described in the previous paragraph, these individual analyses were underpowered.

In summary, the comparison of the psychometric properties of the original and ChatGPT-generated tests showed mostly small effects across all five tests, though statistical equivalence was confirmed only for test difficulty. Differences in clarity fell within the confidence interval for a small effect, while realism ratings were slightly higher for ChatGPT-generated tests, with the lower CI boundary slightly exceeding $d \pm 0.20$. Item content diversity was lower for ChatGPT-generated tests, with a small but notable effect. The correlation between original and ChatGPT-generated tests was strong. For internal consistency, correlations with vocabulary knowledge (StuVoc), and correlations with another ability EI test, the same pattern emerged: no significant differences in moderator tests but no statistical equivalence within the bounds of $r \pm 0.15$. Effect sizes were small, but confidence intervals exceeded the predefined equivalence bounds. However, none of the CI boundaries exceeded a medium effect size ($d \pm 0.50$). Overall, ChatGPT-created tests were largely comparable to the original versions in psychometric properties, with the potential exceptions of slightly lower item content diversity, slightly higher internal consistency, and slightly weaker associations with vocabulary knowledge and other ability EI tests, for which evidence was inconclusive.

## Discussion

Overall, the present study demonstrated that six widely used LLMs (ChatGPT-4, ChatGPT-o1, Copilot 365, Claude 3.5 Haiku, DeepSeek V3, and Gemini 1.5 Flash) outperformed the average human scores on five different ability EI tests, with large effect sizes. At the same time, the moderate-to-high correlation between the proportions of correct responses among humans and the six LLMs across the test items suggests that humans and LLMs may leverage the cues present in the item texts in a similar way to arrive at the correct solutions. Additionally, in the second part of the present project, ChatGPT-4 proved effective in creating situational judgment items to assess the central ability EI domains of emotion knowledge/ understanding and emotion regulation/ management. Across five studies with human participants, original and ChatGPT-generated tests demonstrated statistically equivalent test difficulty. Perceived item clarity and realism, item content diversity, internal consistency, correlations with a vocabulary test, and correlations with an external ability emotional intelligence test were not statistically equivalent between original and ChatGPT-generated tests. However, although some differences slightly exceeded our predefined benchmark for similarity ($d \pm 0.20$), all differences remained below $d \pm 0.25$, and none of the 95% confidence interval boundaries exceeded a medium effect size ($d \pm 0.50$). Additionally, original and ChatGPT-generated tests were strongly correlated ($r = 0.46$). Our findings thus support the idea that ChatGPT can generate responses that are consistent with accurate knowledge of emotional concepts, emotional situations, and their implications.

These results contribute to the growing body of evidence that LLMs like ChatGPT are proficient—at least on par with, or even superior to, many humans—in socio-emotional tasks traditionally considered accessible only to humans, including Theory of Mind[17], describing emotions of fictional characters[23], and expressing empathic concern[18]. These findings have major implications for the use of LLMs in social agents as well as for the assessment of socio-emotional skills.

First, the findings solidify the potential of ChatGPT-4 as a tool for emotionally intelligent interactions. In the context of the debate on whether LLMs and AI can sufficiently convey empathy (e.g. refs. 19,20,44), the results suggest that ChatGPT-4 at least fulfills the aspect of cognitive empathy, meaning its responses are consistent with accurate reasoning about emotions and about how they can be regulated or managed. This capability is crucial for LLMs to function as emotionally intelligent agents that can achieve positive socio-emotional outcomes for users in applied fields such as healthcare (e.g., in socially assistive robots or as mental health chatbots), hospitality, and customer service.

In these settings, LLMs may offer two significant advantages. On the one hand, they process emotional scenarios based on the extensive datasets they have been trained on, whereas humans process them based on their individual knowledge and experience. LLMs may thus have a lower probability of making errors. Although the datasets on which LLMs are trained may partly contain false information, the strong performance in solving ability EI tests in the present study suggests that ChatGPT -4's broad-based reasoning about emotions is generally reliable and aligned with current psychological theories. On the other hand, LLMs can provide consistent application of emotional knowledge, unaffected by the variability typically seen in human emotional performance. Specifically, humans may not always exhibit maximal performance in emotionally charged situations due to factors like mood, fatigue, personal preferences, or competing demands, and research has shown that maximal performance in emotion-related tasks often differs from typical performance (i.e., what people usually do[45]). For example, people sometimes are sometimes deliberately inaccurate when interpreting others' thoughts and feelings ("motivated inaccuracy"[46]). In contrast, AI systems like ChatGPT-4 can reliably deliver maximal performance in emotion understanding and management in every interaction, potentially offering more consistent and effective emotional support.

Although these findings do not address whether AI can simulate affective empathy (i.e., the ability to feel with someone[20]), it is important to note that many AI applications may not require this to achieve their intended outcomes. For example, chatbots or leadership tools designed to

**Table 7 | Realism ratings and number of categories in the sorting task for the original and ChatGPT-created test versions**

| Study | Realism ratings (0–100) | | | | Number of categories in sorting task | | | |
|---|---|---|---|---|---|---|---|---|
| | Original | GPT | t test | Cohen's d [95% CI] | Original | GPT | t test | Cohen's d [95% CI] |
| STEM-B | 86.3 (15.2) | 87.1 (15.9) | t(89) = −0.956, p = 0.342 | −0.101 [−0.308; 0.107] | 5.20 (2.15) | 5.29 (2.37) | t(89) = −0.448, p = 0.655 | −0.047 [−0.254; 0.160] |
| STEU-B | 82.4 (12.3) | **86.5 (10.0)** | t(93) = −6.133, p < 0.001 | −0.633 [−0.853; −0.410] | 7.88 (2.57) | 7.73 (2.64) | t (93) = 0.758, p = 0.563 | 0.078 [−0.125; 0.280] |
| GEMOK-Blends | **81.7 (12.5)** | 75.7 (14.9) | t(90) = 5.508, p < 0.001 | 0.577 [0.354; 0.798] | 7.49 (2.87) | 6.84 (2.59) | t(90) = 2.500, p = 0.023 | 0.262 [0.052; 0.470] |
| GECo Regulation | 73.3 (16.2) | **76.9 (14.7)** | t(94) = −3.812, p < 0.001 | −0.391 [−0.599; −0.181] | **7.41 (2.74)** | 6.27 (2.77) | t(84) = 3.924, p = 0.005 | 0.426 [0.202; 0.647] |
| GECo Management | 74.7 (15.3) | **78.0 (14.0)** | t(96) = −3.342, p = 0.001 | −0.339 [−0.543; −0.134] | **6.36 (2.64)** | 5.84 (2.69) | t(96) = 3.170, p = 0.005 | 0.322 [0.117; 0.525] |
| Pooled dataset | 79.6 (15.2) | 80.8 (14.8) | t(466) = −2.746, p = 0.006 | −0.127 [−0.218; −0.036] | 6.87 (2.77) | 6.44 (2.74) | t(456) = 4.446, p < 0.001 | 0.208 [0.115; 0.301] |

Values in boldface indicate a significantly higher value compared to the other test version. Displayed p values for the individual studies are FDR-corrected.

**Table 8 | Cronbach's alphas and average item-total correlations for the original and ChatGPT-created test versions, and correlations between the original and ChatGPT-created test versions**

| Study | Cronbach's alpha (average item-total correlation) | | | Correlations between original and GPT versionr |
|---|---|---|---|---|
| | Original | GPT | Comparison Cronbach's alpha original/ GPT | |
| STEM-B | 0.48 (0.160) | 0.58 (0.222) | $\chi^2$ = 1.031; df =1; $p$ = 0.388 | 0.35*** |
| STEU-B | **0.41** (0.123) | 0.11 (0.047) | $\chi^2$ = 4.247; df =1; $p$ = 0.098 | 0.42*** |
| GEMOK-Blends | 0.57 (0.191) | 0.43 (0.134) | $\chi^2$ = 1.735; df =1; $p$ = 0.315 | 0.27* |
| GECo Regulation | 0.76 (0.283) | **0.94** (0.579) | $\chi^2$ = 76.224; df =1; $p$ = 0.005 | 0.74*** |
| GECo Management | 0.49 (0.150) | 0.43 (0.127) | $\chi^2$ = 0.326; df =1; $p$ = 0.568 | 0.42*** |
| Weighted mean item-total correlations | 0.18* | 0.26** | QM = 0.635; df = 1; $p$ = 0.426; d = –0.152 [–0.547; 0.223] | Weighted mean r = 0.46*** |

Values in boldface indicate a significantly higher value compared to the other test version. Cronbach's alphas between the original and ChatGPT-created test versions for each study were compared using the R package cochron[57]. Displayed $p$ values for the individual studies are FDR-corrected. For comparing internal consistency between original and GPT-created tests across the five studies, average item-total correlations were computed for each test version and then analyzed with fixed effects multilevel meta-analysis with the restricted maximum likelihood (REML) method in the R package "metafor"[43]. QM is the test statistic for the moderator variable "original" vs "GPT-created" when the intercept is included. Weighted mean r for the correlations between original and GPT-created test versions was obtained using mini meta-analysis[58]. * $p$ < 0.05; ** $p$ < 0.01; *** $p$ < 0.001.

**Table 9 | Correlations for the original and ChatGPT-created test versions with the Stuvoc vocabulary test and with the other ability EI test included in the respective study**

| Study | Correlations with StuVoc | | | Other EI test | Correlations with the other EI test | | |
|---|---|---|---|---|---|---|---|
| | r for original | r for GPT | Comparison original/ GPT | | r for original | r for GPT | Comparison original/ GPT |
| STEM-B | –0.01 | –0.12 | Z = 0.904, $p$ = 0.484 | GECo Management | 0.26* | 0.12 | Z = 1.178, $p$ = 0.600 |
| STEU-B | 0.37*** | 0.28** | Z = 0.958; $p$ = 0.484 | GEMOK-Blends | 0.45*** | 0.38*** | Z = 0.704, $p$ = 0.793 |
| GEMOK-Blends | 0.18 | 0.07 | Z =0.865; $p$ = 0.484 | STEU-B | 0.20 | 0.14 | Z = 0.476, $p$ = 0.793 |
| GECo Regulation | 0.25* | 0.21* | Z =0.549; $p$ = 0.583 | STEM-B | 0.18 | 0.18 | Z = 0, $p$ = 1 |
| GECo Management | 0.39*** | 0.23* | Z = 1.548; $p$ = 0.484 | STEM-B | 0.50*** | 0.33*** | Z = 1.750, $p$ = 0.400 |
| Weighted mean rs | **0.24**** | 0.14 | QM = 2.651; df = 1; $p$ = 0.104; d =0.217 [–0.044; 0.492] | Weighted mean rs | 0.32*** | 0.24*** | QM = 2.189; df = 1; $p$ = 0.149; d =0.197 [–0.064; 0.471] |

The correlations between original and GPT-created tests within each study were compared using Steiger's Z test for dependent correlations[59]. Displayed $p$ values for the individual tests are FDR-corrected. The correlations between original and GPT-created tests across the five studies were compared using fixed effects multilevel meta-analyses conducted with the restricted maximum likelihood (REML) method in the R package "metafor"[43]. QM is the test statistic for the moderator variable "original" vs "GPT-created" when the intercept is included. * $p$ < 0.05; ** $p$ < 0.01; *** $p$ < 0.001.

manage employees' well-being can still support users by providing advice, demonstrating empathic behaviors like active listening[47], and helping users feel heard and understood, regardless of whether the AI actually "feels" empathy[19].

A second important implication of the present research is that LLMs like ChatGPT can be powerful tools for assisting the psychometric development of standardized questionnaires and performance-based assessments, especially in the domain of emotion. Traditionally, developing these tests involves collecting a large number of emotional scenarios through interviews, followed by extensive validation studies[26]. In the present research, ChatGPT-4 was able to generate complete tests with generally acceptable psychometric properties using only few prompts, even for tests with a complex item structure such as the GECo Management test[26], which required response options corresponding to specific conflict management strategies, and the GEMOK-Blends test[25], where scenarios needed to represent blends of emotions as well as various emotional components like action tendencies and physiological expressions. However, it should be noted that the psychometric properties (e.g., test difficulty and Cronbach's alpha) varied between tests. For example, the ChatGPT-generated STEM-B was easier than the original STEM-B, containing many very easy items, while the ChatGPT-generated GEMOK-Blends was more difficult and included several items where only a few test-takers chose the correct response (see Supplementary Notes 5, pp. 48–58, for item-level analyses). In addition, results indicated

that overall, the original tests performed slightly better in construct validity. This could similarly be due to some poorly performing items (e.g., too easy or too difficult items) that do not adequately discriminate between test-takers. These results suggest that while ChatGPT-4 is a valuable tool for generating an initial item pool, it cannot replace the pilot and validation studies needed during test development, which serve to refine or eliminate poorly performing items.

## Limitations

Despite the promising results, several limitations and open questions must be acknowledged. First, this study was conducted using standardized tests with clear and predefined structures, which may not fully capture the complexities of real-world emotional interactions. In natural conversations, emotional scenarios are often ambiguous, incomplete, or require interpretation of subtle cues. There is evidence that LLMs' performance can be disrupted by even minor changes in prompts, suggesting that their ability to handle more complex, less structured emotional tasks may be limited[48]. Further research is needed to assess how ChatGPT and other LLMs compare to humans in understanding and managing emotional situations that are less straightforward, involve conflicting information, or require reading between the lines. Additionally, more research is needed to examine the extent to which LLMs can integrate context and past information from a conversation (e.g., about an individual's personality, preferences, or background information leading to a specific emotional experience), as existing

studies often rely on responses to single prompts rather than longer, more nuanced conversations (e.g. refs. 10,17,23).

Second, the present research was conducted in a Western cultural context, with tests developed in Australia and Switzerland and a training dataset for ChatGPT-4 and the other LLMs that is largely Western-centric. Emotional expressions, display rules, and regulation strategies vary significantly across cultures (e.g. refs. 49,50), meaning that responses deemed correct in a Western context may not be appropriate or effective in other cultural settings[51]. This cultural bias could limit the utility of current LLMs in social and conversational agents designed for non-Western populations[52,53]. Further research is necessary to explore how well LLMs adapt to non-Western cultural contexts and whether they can accurately consider different cultural settings when creating new test items.

Another important limitation is the black box nature of LLMs, where the processes by which the AI arrives at correct answers or generates new items remain unclear (see also the discussion around explainable AI[54]). This lack of transparency makes it difficult to predict how future versions of the model might perform. For example, changes in the model's architecture or training data could result in different, potentially less effective outcomes, such as less creative or diverse scenarios when prompted to create new test items[55]. On the other hand, Kosinski[16] showed that more recent LLMs outperformed older models in solving false-belief ToM tasks, from which he concluded that ToM may increase as a byproduct of newer LLM's improved language skills. The same might apply to ability EI, which would mean that future versions should maintain or increase their performance levels.

## Conclusion
To conclude, while the study reveals some limitations, particularly regarding cultural applicability and the complexity of real-world interactions, the results are encouraging. Six LLMs demonstrated substantial potential in performing EI assessments, and ChatGPT-4 showed notable capability in creating such assessments. These results suggest that LLMs could serve as valuable tools to support socio-emotional outcomes in emotionally sensitive domains, even if they do not fully replicate human affective empathy[19]. At the very least, they can assist users in gaining new perspectives on emotional situations and help them make more informed, emotionally intelligent decisions. This capability positions LLMs such as ChatGPT-4 as a promising resource for enhancing the integration of AI in human-computer interactions and supports the idea that LLMs may be strong candidates for artificial general intelligence (AGI) systems[10].

## Data availability
The data for this research is available on OSF in Microsoft Excel and SPSS format: https://osf.io/mgqre/files/osfstorage.

## Code availability
The code for this research is available on OSF in a text file and can be copied into SPSS syntax or R, respectively. https://osf.io/mgqre/files/osfstorage/67e4b303d7dac4b1728e5a4d[56].

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

## Acknowledgements

We received no specific funding for this work. We would like to thank Joëlle Reinhart, Laura Zimmermann, and Rahel Zubler for their help with data collection.

## Author contributions

K.S. participated in the conceptualization, funding acquisition, and investigation, conducted the formal analysis, wrote the original draft and participated in the review and editing of the manuscript. N.S. participated in the conceptualization, investigation, and review and editing of the manuscript. M.M. participated in the conceptualization, funding acquisition, and review and editing of the manuscript.

## Competing interests

The authors declare no competing interests.
