## [Transparent Peer Review file · Communications Psychology]

Large language models are proficient in solving and creating emotional intelligence tests

Corresponding Author: Dr Katja Schlegel

Version 0:

Decision Letter:

Dear Dr Schlegel,

I apologise that the first email was missing one report. Please find reviewer #2's report in this email below. Thank you.

Thank you for your patience during the peer-review process. Your manuscript titled "Generative AI in action: ChatGPT-4 is proficient in solving and creating emotional intelligence tests" has now been seen by 2 reviewers, whose comments are appended below. I have discussed the reports with my colleagues and I regret to inform you that we decided that in light of the referee reports, we cannot publish your manuscript in Communications Psychology.

You will see that the reviewers raise substantive concerns. Taking these points together with our editorial considerations, these reservations preclude publication of this study in Communications Psychology.

In general, both reviewers were not convinced that the current results provide sufficient evidence to support the central conclusion relating to whether ChatGPT-4 possesses implicit emotional knowledge structure.

Although we cannot offer to publish your manuscript, we believe the Editorial Board at Scientific Reports will find it interesting and recommend you transfer there. To transfer your manuscript there, please use our [manuscript transfer portal](https://mts-commspsychol.nature.com/cgi-bin/main.plex). You will not have to re-supply manuscript metadata and files, unless you wish to make modifications. For more information, please see our [manuscript transfer FAQ](http://www.nature.com/authors/author_resources/transfer_manuscripts.html?WT.mc_id=EMI_NPG_1511_AUTHORTRANSF&WT.ec_id=AUTHOR) page.

I am sorry that we cannot be more positive on this occasion and thank you for the opportunity to consider your work.

Best regards,

Troy Lui

Troy Lui, PhD
Associate Editor
Communications Psychology

REVIEWERS' EXPERTISE:

Reviewer #1: AI/ChatGPT, affective computing
Reviewer #2: AI/ChatGPT, social cognition

REVIEWER COMMENTS:

Reviewer #1 (Remarks to the Author):

Please see attached.

Reviewer #2 (Remarks to the Author):

The manuscript presents a study investigating the emotional intelligence (EI) capabilities of ChatGPT-4, focusing on its performance in solving and generating performance-based test items for emotion understanding and management. ChatGPT-4 outperformed human averages on five standardized EI tests, achieving an accuracy of 86% compared to 56% in human validation samples. Additionally, it generated new test items that were comparable to the originals across several dimensions. The authors conclude that ChatGPT-4 possesses inherent knowledge of emotional concepts and can emulate human-like emotional reasoning, suggesting its potential for emotionally intelligent interactions across various fields.

While the study presents intriguing findings, I have a few concerns:

Validity of Generated Items as Evidence of EI. The rationale for linking item generation to emotional intelligence is unclear to me. The ability of ChatGPT-4 to generate test items comparable to the originals does not inherently demonstrate emotional intelligence. To substantiate this claim, the generated items would need to show meaningful variations not only in linguistic structure but also in the emotional scenarios and specific emotions involved. Without a quantitative analysis of such variations, the new items could simply be rephrased versions of the originals, offering little evidence of advanced emotional reasoning.

Psychometric Comparability Claims. The authors assert that the newly generated items are psychometrically comparable to the original ones, yet this strong claim is not fully supported by the data. Differences in Cronbach's alpha between the two sets, for example, suggest variability in how well the items measure the same underlying construct. Such differences could stem from variations in item difficulty, diversity, or alignment with the intended construct. However, these potential sources of discrepancy are not explored.

Note on appeals: In exceptional circumstances, it is in authors' interest to appeal an editorial decision. More information on appeals is available here: <https://www.nature.com/commmspsychol/submit/editorial-process#appeals>

Version 1:

Decision Letter:

Dear Dr Schlegel,

Thank you for your correspondence asking us to reconsider our decision on your Article, "Generative AI in action: ChatGPT-4 is proficient in solving and creating emotional intelligence tests". After careful consideration and discussion with the Chief Editor, we have decided that we would be willing to consider a full appeal in the form of a revised version of your manuscript.

The referees were concerned about multiple methodological aspects, including whether the evidence could sufficiently support the conclusion that generative AI displays emotional intelligence. Editorially, it is a priority that studies that focus on - in a wider sense - LLM "cognition" also provide an advance that is either of practical relevance or theoretically informative for (human) psychology.

We would be willing to look at a revision that fully addresses the referee concerns for which it is essential but not fully sufficient to demonstrate that ChatGPT can construct new items in the EI generation task that show meaningful variability. Moreover, we ask that the revision provides more comprehensive insights into differences between ChatGPT and humans on the tasks. Finally, given the speed of the field, we ask that you use the newest available ChatGPT in the revised version. You are encouraged to also include additional open-source models.

Along with your revised manuscript, you should also submit a separate point-by-point response to all of the concerns raised by the referees, in each case describing what changes have been made to the manuscript.

Please note that we will only take the appeal forward and contact the reviewers again if we are persuaded that a substantial attempt has been made to address all editorial concerns and referees' comments. In this case, your revised manuscript and the point-by-point reply will be sent back to the referees so that they can judge whether their concerns have been addressed satisfactorily or otherwise.

I should stress, however, that we would be reluctant to trouble our referees again unless we thought that their comments had been addressed in full.

When revising your paper please:

- ensure that it complies with the editorial policies linked below

- ensure it meets our format requirements as set out in our [Guide to Authors](http://www.nature.com/commmspsychol/submit/guide-to-authors).

- ensure that the statistics reporting and interpretation is in line with journal guidelines
<https://www.nature.com/commpsychol/submit/submission-guidelines#statistical-guidelines>

Please mark all correspondence via email with your Communications Psychology reference number in the subject line.

If the revision process takes significantly longer than five months, we will be happy to reconsider your paper at a later date, provided it still presents a significant contribution to the literature at that stage.

Please use the following link to submit your revised manuscript, point-by-point response to the Reviewers' comments with a list of your changes to the manuscript text (which should be in a separate document to any cover letter) and any completed checklist:

Link Redacted

Best regards,

Troy Lui

Troy Lui, PhD
Associate Editor
Communications Psychology

EDITORIAL POLICIES AND FORMATTING

Editorial Policy: [Policy requirements](https://www.nature.com/documents/nr-editorial-policy-checklist.pdf) (Download the link to your computer as a PDF.)

Furthermore, please align your manuscript with our format requirements, which are summarized in our style and formatting guide [Communications Psychology formatting guide](https://www.nature.com/documents/commpsychol-style-formatting-guide-accept.pdf) .

Version 2:

Decision Letter:

Dear Dr Schlegel,

Thank you for your patience during the peer-review process. Your manuscript titled "Emotional artificial intelligence: LLMs are proficient in solving and creating emotional intelligence tests" has now been seen by 2 reviewers, and I include their comments at the end of this message. They find your work of interest but raised some important points. We are interested in the possibility of publishing your study in Communications Psychology, but would like to consider your responses to these concerns and assess a revised manuscript before we make a final decision on publication.

We therefore invite you to revise and resubmit your manuscript, along with a point-by-point response to the reviewers. Please highlight all changes in the manuscript text file.

Editorially, we require you to address methodological concerns and ensure the paper fully complies with Communications Psychology's guidelines and policies. Currently, the manuscript does not satisfy the journal's criteria for transparency and clarity in presentation; appropriate reporting and interpretation of statistics; and style-compliant formatting. These editorial requirements are listed in the attached Request Table; I include additional guidance on some of these requests in the checklist, but all items regardless of whether they contain additional specifications need to be satisfied.

The manuscript currently doesn't state clearly enough what you adopted a priori as the criteria for the inference that LLMs create test items with similar psychometric features as standardized tests. This means it remains ambiguous how the statistics reported in Tables 3-4 inform the question. This needs to be revised (please note our requirements for the interpretation of non-significant differences in NHST as detailed in the checklist).

Please provide a more comprehensive report of your power analysis, including details on the exact effect size you powered for rather than referring to it as "medium" and provide a sensitivity analysis that clarifies the power you achieved for the smallest effect size of interest in the combined data. As this is a common mistake, I highlight that we will require a scientific justification for what effect size was adopted as the smallest effect size of interest, which does not equal the smallest effect size established in the analyses.

In response to the remaining reviewer comments, we highlight that in instances where multiple analyses are used to test the same hypotheses, we expect correction for multiple comparisons. Where the hypotheses are distinct, and especially in the case of preregistration of hypotheses and tests, this requirement will be relaxed. Please also include the Supplementary Tables 8 and 9 in the main text.

I am attaching an Editorial Requests Table that details critical reporting requirements for the revised manuscript. Please attend to each item and ensure your manuscript is fully compliant. If your revised manuscript is not aligned with these requests on major issues, such as those concerning statistics, it may be returned to you for further revisions without re-review.

Please submit the following items:

- Revised manuscript
- Point-by-point response to the referees' comments
- Cover letter (as a separate document)
- [Nature Research Reporting Summary](https://www.nature.com/documents/nr-reporting-summary.zip)
- [Editorial Policy Checklist](https://www.nature.com/documents/nr-editorial-policy-checklist.pdf)
- Completed Editorial Request Table (attached).

via this link: Link Redacted .

Additional guidance is available in our style and formatting guide [Communications Psychology formatting guide](https://www.nature.com/documents/commpsychol-style-formatting-guide-accept.pdf).

Best regards,

Troby Lui

Troby Lui, PhD
Associate Editor
Communications Psychology

REVIEWER EXPERTISE:

Reviewer #1: AI/ChatGPT, affective computing

Reviewer #2: AI/ChatGPT, social cognition

REVIEWER REPORTS:

Reviewer #1 (Remarks to the Author):

Please see attached.

Reviewer #2 (Remarks to the Author):

To substantiate the claim that ChatGPT did not simply rephrase the original items, the authors conducted a similarity analysis in which participants rated the similarity between generated and original items on a 7-point Likert scale (1 = very different, 7 = very similar). They reported that across the five tests, 88% of the ChatGPT-generated scenarios "received similarity ratings below 5 (where 1 = very different and 7 = very similar), indicating that participants generally perceived little to no similarity". However, the authors set an arbitrarily high threshold by defining similarity as only 6 or 7. Given that the scale ranges from 1 to 7, a rating of 5 still suggests moderate similarity, and excluding it may misrepresent the data. A more transparent approach would be to either report the full distribution of ratings or provide a clear rationale for the chosen threshold.

All statistical tests need to be corrected for multiple comparisons to ensure the validity of the findings.

Version 3:

Decision Letter:

Dear Dr Schlegel,

Your manuscript titled "Large language models are proficient in solving and creating emotional intelligence tests" has now been editorially evaluated. I am delighted to say that we are happy, in principle, to publish a suitably revised version in Communications Psychology.

We therefore invite you to revise your paper one last time to address a list of editorial requests. At the same time we ask that you edit your manuscript to comply with our format requirements and to maximise the accessibility and therefore the impact of your work.

EDITORIAL REQUESTS:

SUBMISSION INFORMATION:

OPEN ACCESS:

* DATA AVAILABILITY:

Link Redacted

Best regards,

Troy Lui

Troy Lui, PhD
Associate Editor
Communications Psychology

Responses to Reviewer 1

Abstract

1. A description of the Methods is needed.

RESPONSE: We have now expanded the description of the methods used as much as possible within the 150 words limit. The revised abstract is:

“Large Language Models (LLMs) demonstrate expertise across diverse domains, yet their capacity for emotional intelligence (EI) remains uncertain. This research examined whether LLMs can solve and generate performance-based EI tests. Results showed that ChatGPT-4, ChatGPT-o1, Gemini 1.5 flash, Copilot 365, Claude 3.5 Haiku, and DeepSeek V3 outperformed humans on five standard EI tests, achieving an average accuracy of 81%, compared to the 56% human average reported in the original validation studies. In a second step, ChatGPT-4 generated new test items for each EI test. These new versions and the original tests were administered to human participants across five studies (total N = 467). Overall, original and ChatGPT-generated tests had comparable difficulty and reliability, and participants perceived them as equally clear and realistic. However, original tests exhibited slightly stronger correlations with external EI and vocabulary measures. These findings suggest that LLMs possess inherent knowledge about human emotions and their regulation.”

Introduction

2. Is it really accurate to claim that because the test has “the same psychometric quality as the original test versions,” this proves that “ChatGPT-4 possesses inherent implicit knowledge on the structure and components of emotions”? Isn't it possible that ChatGPT is simply rephrasing names, situations, and words rather than truly possessing "inherent implicit knowledge"? For example, the so-called newly created item, “Nina watched as her younger sibling received a trophy for remarkable achievements in science. a) Jealousy b) Pride c) Contempt d) Excitement e) Surprise,” was merely rephrased to “Nina watched as her younger brother won a trophy for his achievements in science. a) Anger b) Pride c) Fear d) Disgust e) Sadness” This seems more like simple rephrasing rather than evidence of deeper knowledge of emotion.

RESPONSE: We believe that the reviewer may have misread the supplementary material. The first item version (“Nina watched her younger brother...”, p. 14) is not from the original STEU-B test but was already generated by ChatGPT-4. We found the initial set of ChatGPT-4-generated items for the STEU-B to be too easy to solve and subsequently prompted ChatGPT-4 to revise them to increase their difficulty (see prompt on p. 16). This resulted in the item set including the second item version (“Nina watched her younger sibling...”, p. 16, with modified response options). Thus, ChatGPT-4 did not paraphrase an original STEU-B test item.

We recognize that including the entire conversations with ChatGPT in the supplementary material may have made it difficult to identify the final ChatGPT-generated item sets. To address this, we now present the final item sets in a separate section (Part II of the supplementary material).

Unfortunately, copyright restrictions prevented us from including the original five tests in the supplementary material for direct comparison with the newly generated item sets. We acknowledge that this may have contributed to the impression that the new item sets are merely paraphrased versions of the originals.

To mitigate this, the revised article now includes links to where the original test items can be accessed (for the STEU and STEM: <https://www.tandfonline.com/doi/suppl/10.1080/02699931.2017.1414687> ;

for the GEMOK Blends: <https://www.tandfonline.com/doi/suppl/10.1080/02699931.2017.1414687>). For the two GEMOK subtests, the items are not publicly accessible.

To systematically examine the similarity between the original and ChatGPT-generated test items, we now conducted a comprehensive rating study in which 434 Prolific participants assessed the degree of similarity for all combinations of original and ChatGPT-created scenarios across the five tests, resulting in a total of 3,174 scenario pairs. Detailed methods and results for this analysis are provided in the supplementary material (Part III; pp. 40–53).

Overall, for 88% of the ChatGPT-created scenarios across the five tests, participants perceived none or little similarity to any of the original test scenarios (mean rating <5 on a 7-point scale from 1 = scenarios are very different to 7 = scenarios are very similar). For the remaining 12% of the scenarios (mean rating of 5 or higher), a content analysis (see pp. 41-42 and Table S9 in the supplementary material) revealed that while there were similarities in the general setting of the respective original/new item pairs (e.g., in one pair flagged as similar, both items were about a target person giving a presentation at work, while in another pair, both items described a person facing obstacles to meet a deadline), the specific situations and emotions differed. For instance, in one scenario, the target person was described as anxious about presenting, whereas in another, they were annoyed by a critical remark during the presentation. In the other item pair, the target person was afraid of missing a deadline in one scenario but frustrated due to being falsely accused of causing a delay in the other.

These findings indicate that ChatGPT-4 did not merely paraphrase existing test items but instead generated distinct and novel scenarios within the constraints outlined in the prompts (e.g., targeting the same list of emotions as in the original tests; see supplementary material Part I for all prompts). Since the vast majority of the generated items (88%) demonstrated minimal similarity to original scenarios and psychometric analyses revealed only minor differences in quality between original and ChatGPT-4-created tests, we believe that it can be concluded that ChatGPT-4 possesses inherent knowledge of emotional situations, namely, understanding of what causes emotions, how people express and appraise them, and how they can be regulated and managed adaptively.

We now included a summary of this additional study in the Methods section of the main manuscript on p. 26 and refer the reader to the detailed description in the supplementary material (Part III; pp. 40–53).

Methods

3. It's odd that the mean age and standard deviation are identical between the STEM and STEU samples, which are from different groups. This should be doublechecked.

RESPONSE: As shown in Table 7, the mean age and standard deviation are not identical between the STEM-B (39.4 years, SD = 10) and the STEU-B (37.1 years, SD = 10.5), but the standard deviations for age of the STEU-B and GEMOK-Blends samples are identical (10.5). We have double-checked this information, and it is correct.

4. In the 'Assessing ChatGPTs performance in solving EI test items' section: it appears the authors only asked ChatGPT once. The human performance that the authors compare ChatGPT's responses to is based on an average from a large dataset, so why did the authors only ask ChatGPT once? For example, Park & Kim (2024) asked ChatGPT ten times to compare with the data from 30 participants. The authors need to justify their approach and compare it to studies like Park & Kim (2024).

RESPONSE: We initially chose to prompt ChatGPT-4 only once because the EI tests assessed are performance-based instruments with objectively correct and incorrect responses (similar to cognitive ability or knowledge tests), and many studies with similar objectives used a single prompt approach, including Bubeck et al. (2023), Elyoseph et al. (2023), Oh (2025), and Orrù et al. (2023). In contrast, in studies like Park & Kim (2024), LLMs were asked to provide subjective ratings, where responses can be expected to vary due to the absence of definitive correct answers.

Nevertheless, we agree that prompting ChatGPT-4 multiple times would add to the robustness of the findings, as would adding other LLMs. In response to the editor's suggestion, we now included five additional LLMs (ChatGPT-o1 released in December 2024, DeepSeek V3, Gemini 1.5 Flash, Copilot 365, and Claude 3.5 Haiku) to evaluate their performance on ability-based EI tests, and prompted each LLM 10 times to solve each of the five tests. The method section was adjusted accordingly on p. 25. All LLMs demonstrated substantially higher accuracy scores compared to human validation study samples (Tables 1 and 2 in the main manuscript), with ChatGPT-o1 and DeepSeek V3 exceeding the human samples mean scores by more than two standard deviations, reinforcing the reliability of our initial findings and enhancing the validity of the results.

References

Bubeck, S., Chandrasekaran, V., Eldan, R., Gehrke, J., Horvitz, E., Kamar, E., Lee, P., Lee, Y. T., Li, Y., Lundberg, S., Nori, H., Palangi, H., Ribeiro, M. T., & Zhang, Y. (2023). *Sparks of Artificial General Intelligence: Early experiments with GPT-4* (arXiv:2303.12712). arXiv.

<https://doi.org/10.48550/arXiv.2303.12712>

Elyoseph, Z., Hadar-Shoval, D., Asraf, K., & Lvovsky, M. (2023). ChatGPT outperforms humans in emotional awareness evaluations. *Frontiers in Psychology, 14*.

<https://www.frontiersin.org/articles/10.3389/fpsyg.2023.1199058>

Park, C., & Kim, J. (2024). Exploring Affective Representations in Emotional Narratives: An Exploratory Study Comparing ChatGPT and Human Responses. *Cyberpsychology, Behavior, and Social Networking, 27*(10), 736–741. <https://doi.org/10.1089/cyber.2024.0100>

Orrù, G., Piarulli, A., Conversano, C., & Gemignani, A. (2023). Human-like problem-solving abilities in large language models using ChatGPT. *Frontiers in Artificial Intelligence, 6*.

<https://doi.org/10.3389/frai.2023.1199350>

Oh, S. (2025). Evaluating Mathematical Problem-Solving Abilities of Generative AI Models: Performance Analysis of o1-preview and gpt-4o Using the Korean College Scholastic Ability Test. *IEEE Access, 13*, 1227–1235. IEEE Access. <https://doi.org/10.1109/ACCESS.2024.3523703>

5. Outliers: I'm not certain, but it seems unusual to classify outliers as those who 'scored 3 standard deviations or more below the mean on any of the included tests' for this type of scale. In studies using reaction time, exceeding 3SD often implies a lack of attention to the task, but what is the rationale for using this criterion with this type of scale? A justification is needed.

RESPONSE: In online studies, excluding participants with extremely low scores on performance-based tests is a widely accepted approach, as such scores often indicate that participants did not engage with the tasks diligently and may have provided random responses. For instance, in the Situational Test of Emotion Management Brief version (STEM-B), a score three standard deviations below the mean (.27) is near the guessing probability (.25), making it highly unlikely to represent genuine ability. Similar exclusion criteria have been applied in other online studies assessing emotional intelligence or

emotion recognition, as noted in Dael et al. (2022), Israelashvili et al. (2019), and Schlegel & Scherer (2018).

References

Dael, N., Schlegel, K., Weaver, A. E., Ruben, M. A., & Schmid Mast, M. (2022). Validation of a performance measure of broad interpersonal accuracy. *Journal of Research in Personality, 97*, 104182. <https://doi.org/10.1016/j.jrp.2021.104182>

Israelashvili, J., Pauw, L. S., Sauter, D. A., & Fischer, A. H. (2021). Emotion Recognition from Realistic Dynamic Emotional Expressions Cohere with Established Emotion Recognition Tests: A Proof-of-Concept Validation of the Emotional Accuracy Test. *Journal of Intelligence, 9*(2), 25. <https://doi.org/10.3390/jintelligence9020025>

Schlegel, K., & Scherer, K. R. (2018). The nomological network of emotion knowledge and emotion understanding in adults: Evidence from two new performance-based tests. *Cognition and Emotion, 32*(8), 1514–1530. <https://doi.org/10.1080/02699931.2017.1414687>

6. The Prolific study section is confusing. Did the participants take both the original test and the GPT-generated test? If so, this should be clearly stated in the methods section.

RESPONSE: Yes, participants completed both versions of one test. We now revised the methods section under the heading “General procedure of psychometric validation studies” as follows to clarify this (p. 27):

“For each of the five tests, a separate online study was conducted on Prolific.com, where participants completed both the original test and the ChatGPT-generated version. For example, one study involved participants completing the original GEMOK-Blends and the ChatGPT-created GEMOK-Blends, while another study had a different sample complete the original STEU-B and the ChatGPT-generated STEU-B, and so on. Each sample also provided ratings of clarity and realism for both test versions, completed a card-sorting task, and took a vocabulary/crystallized intelligence test, as well as an additional test measuring the same EI dimension as the focal test (see Table 6 and description below). The order of presentation for the original and ChatGPT-generated test versions, along with the associated clarity and realism ratings and the card-sorting task, was randomized. These were followed by the vocabulary test and the other EI test, which were always presented in this fixed order. Participants were not informed that one of the test versions had been generated by ChatGPT, and no references to LLMs or AI were made throughout the study.”

Results

7. Overall, the original and ChatGPT-created versions did not significantly differ in test difficulty.’: While this is true for the pooled dataset, when examined individually, there are noticeable differences. This discrepancy needs to be addressed in the discussion section.

RESPONSE: Indeed, the ChatGPT-created versions were significantly easier for the STEM and GEC_o Regulation subtests (indicated by higher mean scores), while the original versions were easier for the GEMOK-Blends and GEC_o Management subtests. We have added item-level statistics for all original and ChatGPT-generated test versions in the supplementary material (pp. 54–64) to provide more detail on these differences.

For the STEM and GEC_o Regulation subtests, which involve reasoning about resolving negative emotional situations, many correct options appeared to be relatively obvious to participants. These

ChatGPT-generated versions may thus be less effective for discriminating among individuals with higher abilities. However, from the perspective of this study, these results support the notion that ChatGPT “knows” how to handle emotional scenarios effectively.

In contrast, the GEMOK-Blends and GECo Management subtests showed lower scores (i.e., higher difficulty) for the ChatGPT-generated versions. Item-level analysis revealed that this was often due to specific items where the correct response was chosen by very few or no participants (see Tables S12a and S14a). Further analysis showed that this was not typically due to ChatGPT generating completely incorrect answers, but rather because the response options within an item were too similar, or the scenario lacked sufficient detail to clearly indicate the best response (see discussion on p. 54). Improving these items could involve providing more distinct response options, adding detailed cues to scenarios, or replacing the problematic items entirely. To improve the ChatGPT-generated STEM and GECo Regulation tests, items could be made more challenging by creating more plausible distractors or introducing greater complexity in the scenarios and response options.

However, we find it important to emphasize that the goal of this project was not to develop finalized test versions for assessing ability EI. Instead, the aim was to explore whether ChatGPT could, in principle, generate appropriate item sets with minimal and straightforward prompts. The presence of certain poorly performing items in the ChatGPT-generated versions should be viewed within this exploratory context.

In addition to providing item-level characteristics and a discussion of these characteristics using the STEU-B as an example in the supplementary material, we have added the following to the discussion section on p. 16 of the main manuscript:

“However, it should be noted that the psychometric properties (e.g., test difficulty and Cronbach’s alpha) varied between tests. For example, the ChatGPT-generated STEM-B was easier than the original STEM-B, containing many very easy items, while the ChatGPT-generated GEMOK-Blends was more difficult and included several items where the correct response was chosen by only a few test-takers (see Supplementary Material, pp. 54–64, for item-level analyses). In addition, results indicated that overall, the original tests performed slightly better in construct validity. This could similarly be due to some poorly performing items (e.g., too easy or too difficult items) that do not adequately discriminate between test-takers. This suggests that while ChatGPT-4 is a valuable tool for generating an initial item pool, it cannot replace the pilot and validation studies needed during test development, which serve to refine or eliminate poorly performing items.”

Discussion

8. “Additionally, ChatGPT-4 proved effective in creating situational judgment items to assess the main ability EI domains—emotion knowledge/understanding and emotion regulation/management.’ Similar to my comment on the introduction.

While the statement, ‘With just a few prompts, it accomplished what would typically require a much longer process using traditional test development methods, and critically, the psychometric properties of the items generated by ChatGPT-4 were generally comparable to those of the original tests.’ is valid, it does not necessarily support the claim that “These findings support the idea that ChatGPT possesses inherent and accurate knowledge of emotional concepts, emotional situations, and their implications, and that it can generalize this knowledge to novel situations.” Similar to my earlier comment, the authors had ChatGPT create items similar to the existing ones, but they didn’t make it apply to different situations or replace it with a similar but different task. If the authors want to make such claims, they should include qualitative analyses along with the quantitative ones. For example, they could provide example items that support their argument by comparing the “original” and “altered” items. It seems the authors are overinterpreting their findings.

RESPONSE: We believe that the similarity rating study described in response to the reviewer's first comment effectively addresses the concern regarding whether ChatGPT generated genuinely new scenarios and responses. To clarify further, the newly generated items were not "altered" versions of the original items, meaning that ChatGPT-4 was not tasked with modifying individual items or creating "parallel" versions one at a time. Instead, ChatGPT-4 was instructed to generate entirely novel item sets for each test in a single session (see supplementary material, Part I).

With the similarity rating study and the item content analysis clearly showing that the newly generated scenarios were distinct from the original tests, we believe that our conclusion regarding the inherent emotion knowledge of ChatGPT-4 is justified.

Responses to Reviewer 2

1. The manuscript presents a study investigating the emotional intelligence (EI) capabilities of ChatGPT-4, focusing on its performance in solving and generating performance-based test items for emotion understanding and management. ChatGPT-4 outperformed human averages on five standardized EI tests, achieving an accuracy of 86% compared to 56% in human validation samples. Additionally, it generated new test items that were comparable to the originals across several dimensions. The authors conclude that ChatGPT-4 possesses inherent knowledge of emotional concepts and can emulate human-like emotional reasoning, suggesting its potential for emotionally intelligent interactions across various fields.

While the study presents intriguing findings, I have a few concerns:

Validity of Generated Items as Evidence of EI. The rationale for linking item generation to emotional intelligence is unclear to me. The ability of ChatGPT-4 to generate test items comparable to the originals does not inherently demonstrate emotional intelligence. To substantiate this claim, the generated items would need to show meaningful variations not only in linguistic structure but also in the emotional scenarios and specific emotions involved. Without a quantitative analysis of such variations, the new items could simply be rephrased versions of the originals, offering little evidence of advanced emotional reasoning.

RESPONSE: We agree that without a detailed comparison of the item contents between the original and ChatGPT-created versions, it could be assumed that the newly generated items are merely rephrased versions of the originals, which would indeed offer little evidence of advanced emotional reasoning.

First, we would like to clarify that ChatGPT-4 was not tasked with modifying individual items or creating "parallel" versions one at a time. Instead, ChatGPT-4 was instructed to generate entirely novel item sets for each test in a single session (see supplementary material for the prompts, Part I), making it less likely that it would simply paraphrase the existing item sets.

Second, to systematically examine the similarity between the original and ChatGPT-generated test items, we now conducted a comprehensive rating study in which 434 Prolific participants assessed the degree of similarity for all combinations of original and ChatGPT-created scenarios across the five tests, resulting in a total of 3,174 scenario pairs. Detailed methods and results for this analysis are provided in the supplementary material (Part III; pp. 40–53), and a summary of the study is now included in the main manuscript on p. 25.

Overall, for 88% of the ChatGPT-created scenarios across the five tests, participants perceived none or little similarity to any of the original test scenarios (mean rating <5 on a 7-point scale from 1 = scenarios are very different to 7 = scenarios are very similar). For the remaining 12% of the scenarios (mean rating of 5 or higher), a content analysis (see pp. 41-42 and Table S9 in the supplementary material) revealed that while there were similarities in the general setting of the respective original/new item pairs (e.g., in one pair flagged as similar, both items were about a target person giving a presentation at work, while in another pair, both items described a person facing obstacles to meet a deadline), the specific situations and emotions differed. For instance, in one scenario, the target person was described as anxious about presenting, whereas in another, they were annoyed by a critical remark during the presentation. In the other item pair, the target person was afraid of missing a deadline in one scenario but frustrated due to being falsely accused of causing a delay in the other. For details on all “similar” item pairs, see Table S9.

The results of the similarity rating study indicate that ChatGPT-4 did not merely paraphrase existing test items but instead generated distinct and novel scenarios within the constraints outlined in the prompts (e.g., targeting the same list of emotions as in the original tests; see supplementary material Part I for all prompts). Since the vast majority of the generated items (88%) demonstrated minimal similarity to original scenarios and psychometric analyses revealed only minor differences in quality between original and ChatGPT-4-created tests, we believe that it can be concluded that ChatGPT-4 possesses inherent knowledge of emotional situations (i.e., understanding of what causes emotions, how people express and appraise them, and how they can be regulated and managed adaptively).

We now included a summary of this additional study in the Methods section of the main manuscript on p. 25 and refer the reader to the detailed description in the supplementary material (Part III; pp. 40–53).

2. **Psychometric Comparability Claims.** The authors assert that the newly generated items are psychometrically comparable to the original ones, yet this strong claim is not fully supported by the data. Differences in Cronbach's alpha between the two sets, for example, suggest variability in how well the items measure the same underlying construct. Such differences could stem from variations in item difficulty, diversity, or alignment with the intended construct. However, these potential sources of discrepancy are not explored.

RESPONSE: Thank you for this comment. We completely agree that the analyses we conducted do not allow us to make claims about the comparability of the newly generated and original items at the item level. Rather, our findings and conclusions refer to test-level properties across the five tests and are supported by small or very small effect sizes when comparing item difficulty, item clarity, realism, diversity (categories in the card-sorting task), Cronbach's alpha, and correlations with other tests. We agree, however, that more detailed comparisons for the individual tests, as well as item-level analyses, would provide additional useful insights.

We have therefore revised the manuscript as follows: First, we carefully reviewed the text to ensure all claims refer to the test level and adjusted language where necessary, including the sentence on p. 15: “With just a few prompts, it accomplished what would typically require a much longer process using traditional test development methods, and critically, the psychometric properties of the items generated by ChatGPT-4 were generally comparable to those of the original tests.” This was revised to: “... and critically, the psychometric properties of the ChatGPT-generated and original tests were generally comparable.”

Second, we added a more detailed analysis of individual tests to the supplementary material (section IV, Tables S10-14), to which we refer on p. 9 of the main manuscript. These tables include item-level mean scores, item-total correlations, and clarity and realism ratings for both test versions, enabling readers to compare the original and ChatGPT-generated versions and explore potential reasons for differences.

Regarding the reviewer's comment on the differences in Cronbach's alpha between the ChatGPT-generated and original test versions, we conducted a closer analysis of the STEU-B (Situational Test of Emotion Understanding). This was the only test where the ChatGPT-generated version exhibited a significantly lower Cronbach's alpha compared to the original version.

At the test-level (as detailed in Tables 3–6 of the main manuscript), there was no apparent reason for the lower alpha in the ChatGPT-generated version, as there was no substantial difference in mean difficulty, content diversity, and correlations with other tests. Clarity and realism ratings were even higher in the ChatGPT-generated version. However, at the item level, four items (4, 7, 8, and 13) showed negative item-total correlations of .10 or higher, contributing to the low alpha (see Table S10a). Among these, three items (7, 8, and 13) had low mean scores (.48, .10, .30), and for one additional item (18), no participant selected the response deemed correct by ChatGPT.

Examining the item texts and response options revealed potential issues. For items 7, 8, and 13, the five response options were relatively similar within each item. For example, in item 13 (about a runner crossing the finish line), the options included excitement, joy, pride, relief, and surprise, where multiple options (or a blend of emotions) could theoretically be considered correct without additional contextual details in the item text. If further test development is pursued, these items could be improved by providing more distinct response options, adding more detailed cues to the scenarios, replacing the items entirely, or adopting consensus scoring (as used in the original STEU) by designating the most frequently chosen option as correct.

To illustrate the potential impact of this last approach, we recoded items 7, 8, 13, and 18 to mark the most frequently chosen option as correct. For item 7, a secondary option was almost as frequently selected (mean scores: .47 versus .48) and was also recoded as correct. Following these adjustments, the item-total correlations for items 7, 13, and 18 became positive, raising Cronbach's alpha to .35. While still low by conventional standards, this value approaches the .41 of the original STEU-B. However, item 8 continued to exhibit a negative item-total correlation, indicating that it would likely be replaced in future test versions.

We have added this discussion on the potential sources of the Cronbach's alpha differences between the two STEU versions to the supplementary material in Section IV.

To provide perspective on the presence of certain poorly performing items in the ChatGPT-generated test versions, we would like to highlight the following: First, the original tests also contained some poorly performing items (e.g., items with negative item-total correlations). Second, and more importantly, the goal of this project was not to develop final test versions for assessing ability EI. Instead, the aim was to evaluate whether ChatGPT could, in principle, generate appropriate item sets without any further modifications. Specifically, we sought to assess the psychometric quality of the full tests, rather than that of individual items, as a means of evaluating ChatGPT's emotion knowledge. This approach—examining the psychometric quality of ChatGPT's initial item pool and comparing it to item pools of published tests that underwent multiple construction phases starting from a larger pool that was iteratively revised and refined—is a particularly stringent test. Under this strict framework, we believe it is remarkable that the ChatGPT-generated versions achieved overall psychometric properties comparable to those of established tests.

To avoid any confusion, we now state very clearly in the discussion (p. 18) that “[..] while ChatGPT-4 is a valuable tool for generating an initial item pool, it cannot replace the pilot and validation studies needed during test development, which serve to refine or eliminate poorly performing items.”

Responses to Reviewer 1

Abstract

1. It's odd that the mean age and standard deviation are identical between the STEM and STEU samples, which are from different groups. This should be doublechecked.

“The STEM was validated in a sample of 112 undergraduate students in Australia (68% female, age $M = 21.1$, $SD = 5.6$; Study 1 in MacCann & Roberts, 2008). The STEU was validated in a sample of 200 undergraduate students in Australia (68% female, age $M = 21.1$, $SD = 5.6$; Study 1 in MacCann & Roberts, 2008).”

RESPONSE: Thank you for the clarification. The STEM sample ($N = 112$) was a subset of the STEU sample ($N = 200$) in Study 1 from MacCann & Roberts (2008). As the authors did not report gender and age separately for the subsample, we had decided to report the same age and gender statistics for the subsample. As we agree that this may be confusing, we now changed this text as follows: “The STEM was validated using a sample of 112 undergraduate students in Australia (Study 1 in MacCann & Roberts, 2008). This sample was a subset of the STEU validation sample ($N = 200$) described below.” (p. 21)

Responses to Reviewer 2

1. To substantiate the claim that ChatGPT did not simply rephrase the original items, the authors conducted a similarity analysis in which participants rated the similarity between generated and original items on a 7-point Likert scale (1 = very different, 7 = very similar). They reported that across the five tests, 88% of the ChatGPT-generated scenarios “received similarity ratings below 5 (where 1 = very different and 7 = very similar), indicating that participants generally perceived little to no similarity”. However, the authors set an arbitrarily high threshold by defining similarity as only 6 or 7. Given that the scale ranges from 1 to 7, a rating of 5 still suggests moderate similarity, and excluding it may misrepresent the data. A more transparent approach would be to either report the full distribution of ratings or provide a clear rationale for the chosen threshold.

RESPONSE: We have now added the following text in the main manuscript to be more transparent (p. 28): “For each ChatGPT-created scenario (20 for GEMOK, 20 for GEMo Emotion Management, 28 for GEMo Regulation, 19 for STEU, and 18 for STEM, totaling 105 scenarios), we identified the original test scenario with the highest perceived similarity. For example, for item 1 from the ChatGPT-created STEU, the most similar original scenario was scenario 36, with a similarity rating of 4.4 (see column 1 in Supplementary Table 2). Table 1 [now added to the main manuscript; see below] presents the distribution of these highest similarity ratings across the 105 ChatGPT-created scenarios.” As suggested by the editor, we have also moved Supplementary Table 9, which displays the item texts for all pairs with similarity ratings above 5.0, to the main manuscript (now Table 2). For completeness, the full similarity ratings for all 3,174 item comparisons remain available in Supplementary Tables 1–8.

Table 1

Distribution of highest similarity ratings for each of the 105 ChatGPT-generated scenarios

Highest similarity rating	frequency	%	Cumulative %
1.0 – 2.0	1	1.0	1.0
2.1 – 3.0	23	21.9	22.9
3.1 – 4.0	36	34.3	57.1
4.1 – 5.0	32	30.5	87.6
5.1 – 6.0	9	8.6	96.2
6.1 – 7.0	4	3.8	100.0

Note. For each newly generated scenario, the value included in the Table represents the highest similarity rating observed across all comparisons with original test scenarios.

2. All statistical tests need to be corrected for multiple comparisons to ensure the validity of the findings.

RESPONSE: We did not apply a correction for multiple comparisons for the analyses on the pooled dataset, because the assessed outcome variables captured largely independent constructs (e.g., test difficulty was unrelated to clarity ratings), and because the hypotheses for all variables were preregistered (see p. 10). For the analyses of the individual EI tests/ separate studies (five tests per outcome variable), we now corrected the p-values for each outcome variable for multiple comparisons using the False Discovery Rate (FDR) correction (Benjamini & Hochberg, 1995; see p. 10)